# AN ISOTROPIC GAUSSIAN PERSPECTIVE ON FULLY CONNECTED LAYERS

## ABSTRACT

We investigate the possibility of representing a fully connected layer in neural networks from a simple yet expressive statistical form, especially modeling it with Isotropic Gaussian distributions parameterized by a minimal set of means and variances. This formulation not only provides a new lens for understanding the statistical structure of fully connected layers but also allows us to reduce the memory requirements of their weight parameters from $\mathcal{O}(n^2)$ to $\mathcal{O}(n)$ in practice. To learn the Isotropic Gaussian distribution of fully connected layers, we propose two distribution learning methods, i.e., Gradient Distribution Update (GDU) and Weight Distribution Update (WDU), which integrate seamlessly with conventional back-propagation. These methods iteratively estimate the best Isotropic mean and variance while ensuring compatibility with other layers and maintaining comparable model performance. Once learned, the fully connected layer is executed by sampling weight parameters from the Isotropic distribution at run-time. The experiments show that fully connected layers can be effectively represented in Isotropic Gaussians, implying the potential of statistical interpretation of widely used neural network operations. The code is available on an anonymous GitHub[1].

## 1 INTRODUCTION

One of the most fundamental components of deep learning architectures is the fully connected layer, also known as the dense layer (Bengio et al., 2017). Recent studies have demonstrated its critical role in tasks such as Transformers (Vaswani, 2017), enhancing image classification performance (Basha et al., 2020; Nerantzis et al., 2022), and in scenarios with limited data (Kocsis et al., 2022). Despite its strong expressive capability, the fully connected layer often imposes substantial constraints in terms of memory size and computational efficiency. This inefficiency becomes particularly pronounced in modern architectures, such as Transformers (Vaswani, 2017), where the computation grows rapidly with increasing token length. For example, GPT-4 (Achiam et al., 2023) requires storing $67M$ weight parameters for a single fully connected layer in its attention, as key, query, and value, when the token length is $n = 8,192$. In this context, the primary objective of this work is to explore a fundamental question: "Is it possible to efficiently represent the weight parameters of a fully connected layer using a simple yet expressive statistical form, and to what extent can this be pushed?" As a preliminary answer to this question, we reinterpret the weight matrix in fully connected layers from an Isotropic Gaussian perspective, investigating its possibilities and limitations.

Given a fully connected layer, we hypothesize that its weight parameter matrix can be effectively approximated by Isotropic Gaussians (Hastie et al., 2009) applied to its rows and columns, respectively. From this, a substantial number of fully connected layers in a neural network can be solely represented with only two parameters of an Isotropic Gaussian distribution, i.e., the mean and variance. This simplified Isotropic Gaussian representation not only allows for more statistical interpretation of fully connected layers but also significantly decreases their memory usage. By retaining only two parameters (i.e., Isotropic Gaussian mean and variance) for each row and column of the weight matrix $\mathbf{W} \in \mathbb{R}^{n \times n}$, instead of storing the entire weight parameters of $n^2$ (Gallicchio et al., 2018; Scardapane & Wang, 2017), its memory usage can be effectively reduced from $\mathcal{O}(n^2)$ to $\mathcal{O}(n)$.

To implement this hypothesis, we introduce two Isotropic Gaussian learning methods, i.e., Gradient Distribution Update (GDU) and Weight Distribution Update (WDU). These methods learn the

---

[1]https://anonymous.4open.science/r/isotropic_gaussian_learning

Isotropic Gaussian mean and variance of the weight matrix in a fully connected layer during standard backpropagation (Figure 1). Once the mean and variance are learned, the fully connected layer is executed by randomly sampling weight parameters from the learned Isotropic Gaussian distribution. Through extensive empirical analysis, we find representing the weight matrices of a selected subset of fully connected layers with Isotropic Gaussians can preserve model performance while substantially reducing memory usage compared to conventional full weight matrix parameterization.

The experimental results show that the Isotropic Gaussian distribution of a fully connected layer can be effectively learned and sampled for its execution. When selectively applied to a subset of fully connected layers, it achieves model performance comparable to that obtained using the entire weight parameters. For example, applying Isotropic Gaussians to ViT (Dosovitskiy et al., 2020) for image classification (Krizhevsky et al., 2012) yields in test accuracies of 76.77% on mini-ImageNet (Vinyals et al., 2016) and 83.98% on Food-101 (Bossard et al., 2014), compared to the baseline 78.46% and 84.67%. For text classification (Zhang et al., 2010) conducted using BERT (Devlin et al., 2019) across various datasets (Zhang et al., 2015; Saravia et al., 2018) demonstrate performance comparable to the baseline models, e.g., on AG News, the accuracy decreases slightly from 94.79% to 94.52%. At the same time, the results also indicate the limits of representing a large set of fully connected layers using solely Isotropic Gaussian distributions. When Isotropic Gaussians are applied more aggressively to a substantial number of layers, model performance tends to degrade proportionally. These results suggest the possibility of representing fully connected layers using Isotropic Gaussians, as well as the boundaries of their expressivity.

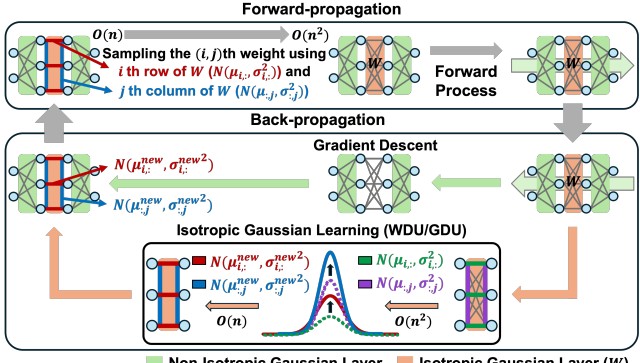

Figure 1: An overview of Isotropic Gaussian learning. Orange indicates layers where Isotropic Gaussian learning is applied, while light green shows non-Isotropic Gaussian layers. The Isotropic mean $\mu$ and variance $\sigma$ of each row and column are updated (from green to red, and from purple to blue, respectively) during backpropagation along with other layers, using either GDU or WDU (details in Section 3).

This study reveals that the weight matrix of a fully connected layer can be effectively and meaningfully represented using a statistical formulation, rather than merely as a collection of individual parameters. Beyond its practical implications for model compression, the proposed Isotropic Gaussian perspective offers new insights into the learning dynamics and statistical structure of fully connected operations, providing a new analytical lens for both theoretical understanding and real applications.

## 2 REPRESENTING WEIGHT MATRIX VIA ISOTROPIC GAUSSIANS

In a fully connected network, each neuron is fully connected to all neurons in the previous layer of the network, i.e., each neuron in the $l$th layer is connected to all neurons in the $(l-1)$th layer. Let $\mathbf{h}^{(l-1)}$ denote the output of the $(l-1)$th layer, with $\mathbf{W}^{(l)}$ and $\mathbf{b}^{(l)}$ representing the weight matrix and bias of the $l$th layer, respectively. Then, the output of the $l$th layer becomes:

$$h_i^{(l)} = f(\sum_{j=1}^{m} w_{i,j}^{(l)} h_j^{l-1} + b_i^{(l)}) \text{ for } i \in \{1, 2, \ldots, n\}. \tag{1}$$

where $n$ and $m$ is the number of neurons in the $l$th and $(l-1)$th layer, respectively, $w_{i,j}^{(l)}$ is the $(i,j)$th element of $\mathbf{W}^{(l)}$, and $f(\cdot)$ is the non-linear activation such as ReLU (Agarap, 2018).

### 2.1 MATRIX-WISE ISOTROPIC GAUSSIAN

Given the weight matrix $\mathbf{W} \in \mathbb{R}^{n \times m}$ in Eq. (1), one may first consider representing the entire weight matrix $\mathbf{W}$ with a single Isotropic Gaussian using a single mean and variance in a matrix-wise manner. Consider a fully connected layer in Eq. (1) with the input $\mathbf{x} \in \mathbb{R}^m = [x_1, \ldots, x_m]$, the output $\mathbf{y} \in \mathbb{R}^n = [y_1, \ldots, y_n]$, the corresponding target $\hat{\mathbf{y}} \in \mathbb{R}^n = [\hat{y}_1, \ldots, \hat{y}_n]$, and the

bias $\mathbf{b} \in \mathbb{R}^n$. Given the output $\mathbf{y} = \mathbf{W}\mathbf{x}^{(p)} + \mathbf{b}$, since the target is $\hat{\mathbf{y}}$, if we optimize $\mathbf{W}$, we get $\mathbf{W} = (\hat{\mathbf{y}} - \mathbf{b})(\mathbf{x}^{(p)})^{-1}$. However, since the inverse of a vector $(\mathbf{x}^{(p)})^{-1}$ is not generally defined, it is challenging to obtain an exact $\mathbf{W}$. Although approximation methods can be used to estimate $\mathbf{W}$, they are inherently inexact and can lead to degradation in model performance. Moreover, representing the $mn$ weight parameters using only a single Isotropic Gaussian mean and variance forces parameters serving different functions to conform to a common statistical pattern. This constraint severely limits the representational capacity of models employing matrix-wise Isotropic Gaussians for fully connected layers. Consequently, modeling the distribution of the entire weight matrix with a single Isotropic Gaussian in a matrix-wise manner is both unsuitable and unstable.

## 2.2 ROW- AND COLUMN-WISE ISOTROPIC GAUSSIAN

As discussed above, the matrix-wise Isotropic Gaussian imposes common statistical constraints on the entire weight matrix, severely limiting the expressive power of fully connected layers as well as the entire model. To address this limitation, we propose using a row- and column-wise Isotropic Gaussian distribution, which is functionally more effective and significantly more expressive.

**Row-wise Isotropic Gaussian distribution.** As shown in Eq. (1), each row $\mathbf{W}_{i,:}^{(l)}$ of the weight matrix $\mathbf{W}^{(l)}$ determines how a neuron $h_i^{(l)}$ in the $l$th layer interacts with $\mathbf{h}^{(l-1)}$ from the previous layer. In other words, $\mathbf{W}_{i,:}^{(l)}$ encapsulates the contribution of all neurons in the previous layer into $h_i^{(l)}$. Thus, distributions of each row in the weight matrix in fully connected layer influence the learning process of the neurons, which, in turn, has a substantial impact on the network's overall output. Therefore, the $i$th row of the weight matrix is applied independently to the $i$th output value of the previous layer. This allows each row of the weight matrix to be regarded as an independent functional unit, implying that each row can possess its own unique statistical characteristics. From this, we propose a more flexible approach that assumes each row $\mathbf{W}_{i,:}$ is a set of samples drawn from a distinct Isotropic Gaussian distribution $\mathcal{N}(\mu_{i,:}, \sigma_{i,:}^2)$ with its own statistical pattern. This row-wise approach reduces the number of parameters from $m$ to just two for each row of the weight matrix, lowering the memory complexity of the entire weight matrix $\mathbf{W}$ from $\mathcal{O}(nm)$ to $\mathcal{O}(n)$.

**Column-wise Isotropic Gaussian distribution.** While the row-wise Isotropic Gaussian can represent the weight matrix distribution of a fully connected layer more effectively than the matrix-wise Isotropic Gaussian, we further enhance its expressiveness by incorporating the statistical patterns of the weight matrix columns into rows, which provides a richer, dual-perspective representation of the weight matrix distribution. These two row- and column-wise distributions offer complementary interpretations of an individual weight $w_{i,j}$. From the row-wise perspective, the $i$th row represents a 'recipe' for how all inputs $\mathbf{x}$ are combined to produce a single output $y_i$. From the column-wise perspective, the $j$th column describes how a single input $x_j$ influences all outputs $\mathbf{y}$.

**Combining row- and column-wise distributions.** To synthesize these two views (i.e., row- and column-wise Isotropic Gaussians), we model the distribution for each individual weight $w_{i,j}$ as follows. Since each weight parameter $w_{i,j}$ is shared between the two Isotropic Gaussian distributions (i.e., row and column), they are not fully independent; rather, they provide different statistical views of the same weight parameter. We model this relationship by assuming $w_{i,j} \sim \mathcal{N}(\mu_{i,j}, \sigma_{i,j}^2)$ and compute the mean $\mu_{i,j}$ and variance $\sigma_{i,j}^2$ of the $i$-th row and $j$-th column of the weight matrix $\mathbf{W}$ as:

$$\mu_{i,j} = \frac{m\mu_{i,:} + n\mu_{:,j}}{m+n}, \quad \sigma_{i,j}^2 = \frac{1}{m+n}((m-1)\sigma_{i,:}^2 + (n-1)\sigma_{:,j}^2) + \frac{mn}{(m+n)^2}(\mu_{i,:} - \mu_{:,j})^2 \quad (2)$$

where $\mu_{i,:}, \sigma_{i,:}^2$ denote the mean and variance from the distribution of $i$th row, and $\mu_{:,j}, \sigma_{:,j}^2$ denote the mean and variance from the distribution of $j$th column. As shown in Eq. (2), the mean $\mu_{i,j}$ is equivalent to a weighted average of the row and column means. Crucially, the variance $\sigma_{i,j}^2$ is composed of two terms: a weighted average of the intrinsic variances and a discrepancy penalty proportional to $(\mu_{i,:} - \mu_{:,j})^2$. This second term serves as an adaptive regularization mechanism, explicitly capturing uncertainty when the two distributions diverge. This approach requires learning $2n$ distribution parameters (i.e., Isotropic Gaussian mean and variance) for the rows and $2m$ for the columns, respectively, resulting in a total of $2n + 2m$ parameters. Consequently, the memory complexity of the entire weight matrix $\mathbf{W}$ is reduced from $\mathcal{O}(nm)$ to $\mathcal{O}(n+m)$, while achieving a richer statistical representation by combining row- and column-wise Isotropic Gaussian distributions.

## 3 ISOTROPIC GAUSSIAN LEARNING

Deriving the mean and variance of Isotropic Gaussian distributions analytically is highly challenging and often requires unrealistic assumptions. To find the mean and variance of the Isotropic Gaussians for each $i$-th row ($\mu_{i,:}, \sigma_{i,:}^2$) and $j$-th column ($\mu_{:,j}, \sigma_{:,j}^2$) of the weight matrix $\mathbf{W}$, we propose two distribution learning methods: (1) Gradient Distribution Update (GDU) and (2) Weight Distribution Update (WDU). In both methods, the mean and variance are iteratively updated during model training, using different ranges of variance, while other layers are trained via standard backpropagation (Kelley, 1960; Rumelhart et al., 1986). This joint optimization allows the Isotropic Gaussian layers to adapt to non-Isotropic layers, and vice versa, enabling them to be represented effectively through Isotropic Gaussians. Naturally integrated into the training procedure, the proposed GDU and WDU enable seamless learning of Isotropic Gaussian representations of fully connected layers.

During forward propagation in model training, weight parameters are randomly sampled from corresponding Isotropic Gaussians using their mean and variance in Eq. (2). In backpropagation (Rumelhart et al., 1986), the mean and variance are updated by GDU or WDU, as described below.

### 3.1 GRADIENT DISTRIBUTION UPDATE (GDU)

Based on backpropagation (Rumelhart et al., 1986), Gradient Distribution Update (GDU) learns the *distribution of weight gradients*. With the gradient descent (Lemaréchal, 2012), $\mathbf{W}$ is updated as:

$$\mathbf{W}^{\text{new}} = \mathbf{W} - \eta \nabla \mathbf{W} \tag{3}$$

where $\eta$ is the learning rate, and $\nabla \mathbf{W} = \frac{\partial \mathcal{L}}{\partial \mathbf{W}}$ is the gradient of the task objective (loss) $\mathcal{L}$ with respect to $\mathbf{W}$. Let $\mathcal{W}_{i,:}$ be the random variable of all weight parameters of the $i$th row of the weight matrix $\mathbf{W}_{i,:}$, $\mathcal{W}_{:,j}$ be the random variable of all weight parameters of the $j$th column of the weight matrix $\mathbf{W}_{:,j}$, and $\mathcal{W}_{i,j}$ be the random variable for the weight at the $(i, j)$th position of the weight matrix $\mathbf{W}_{:,j}$, obtained by Eq. (2). For the forward propagation of the network, we randomly sample $w_{i,j}$ from the current Isotropic Gaussian $\mathcal{W}_{i,j} \sim \mathcal{N}(\mu_{i,j}, \sigma_{i,j}^2)$, instead of maintaining each $w_{i,j}$.

In GDU, the distribution of the weight gradient $\nabla \mathcal{W}_{i,:}$ is derived from the gradient of the sampled weight matrix $\nabla \mathbf{W}_{i,:}$. Then, $\nabla \mathcal{W}_{i,:}$ is applied to $\mathcal{W}_{i,:}$ to obtain $\mathcal{W}_{i,:}^{\text{new}}$ in a similar manner in Eq. (3).

$$\mathcal{W}_{i,:}^{\text{new}} = \mathcal{W}_{i,:} - \eta \nabla \mathcal{W}_{i,:} \tag{4}$$

While it is straightforward to compute Eq. (3) involving the weight matrix $\mathbf{W}$ consisting of a set of values, it is not easy for the distribution Eq. (4) involving updating the distribution of $\mathcal{W}_{i,:}$, i.e., updating its mean and variance, which necessitates a distributional approach. Since Eq. (4) is the sum of two random variables, i.e., $\mathcal{W}_{i,:}$ and $-\eta \nabla \mathcal{W}_{i,:}$, the mean $\mu_{i,:}^{\text{new}}$ and variance $(\sigma_{i,:}^{\text{new}})^2$ of $\mathcal{W}_{i,:}^{\text{new}}$ can be calculated by:

$$\mu_{i,:}^{\text{new}} = \mathbb{E}[\mathcal{W}_{i,:}^{\text{new}}] = \mathbb{E}[\mathcal{W}_{i,:}] + \mathbb{E}[-\eta \nabla \mathcal{W}_{i,:}] \tag{5}$$

$$(\sigma_{i,:}^{\text{new}})^2 = \text{Var}(\mathcal{W}_{i,:}^{\text{new}}) = \text{Var}(\mathcal{W}_{i,:}) + \text{Var}(-\eta \nabla \mathcal{W}_{i,:}) + 2\text{Cov}(\mathcal{W}_{i,:}, -\nabla \mathcal{W}_{i,:})$$
$$= \text{Var}(\mathcal{W}_{i,:}) + \text{Var}(-\eta \nabla \mathcal{W}_{i,:}) + 2\rho_{i,:}\text{Std}(\mathcal{W}_{i,:})\text{Std}(-\eta \nabla \mathcal{W}_{i,:}) \tag{6}$$

where $\mathbb{E}[\cdot]$ and $\text{Var}(\cdot)$ computes the expectation (mean) and variance of the elements in the vector or matrix, respectively, $\text{Cov}(\mathcal{W}_{i,:}, \nabla \mathcal{W}_{i,:})$ denotes the covariance between $\mathcal{W}_{i,:}$ and $\nabla \mathcal{W}_{i,:}$, $\rho_{i,:} := \text{corr}(\mathcal{W}_{i,:}, -\eta \nabla \mathcal{W}_{i,:}) = \frac{\mathbb{E}[(\mathcal{W}_{i,:} - \mu_{i,:})(-\eta \nabla \mathcal{W}_{i,:} - \mathbb{E}[-\eta \nabla \mathcal{W}_{i,:}])]}{\sigma_{\mathcal{W}_{i,:}} \sigma_{-\eta \nabla \mathcal{W}_{i,:}}}$ is the correlation between $\mathcal{W}_{i,:}$, and $-\eta \nabla \mathcal{W}_{i,:}$. From Eq. (5) and (6), we have the updated row-wise Isotropic Gaussian $\mathcal{W}_{i,:}^{\text{new}} \sim \mathcal{N}(\mu_{i,:}^{\text{new}}, (\sigma_{i,:}^{\text{new}})^2)$. Similarly, an updated column-wise Isotropic Gaussian $\mathcal{W}_{:,j}^{\text{new}} \sim \mathcal{N}(\mu_{:,j}^{\text{new}}, (\sigma_{:,j}^{\text{new}})^2)$ can be obtained. Algorithm 1 in Section C describes the procedure of GDU.

Since $\mathcal{W}_{i,:}^{\text{new}}$ and $\mathcal{W}_{:,j}^{\text{new}}$ are updated based on the distributions of the weight gradient $\nabla \mathcal{W}_{i,:}$ and $\nabla \mathcal{W}_{:,j}$ as in Eq. (6), the variances $(\sigma_{i,:}^{\text{new}})^2 = \text{Var}(\mathcal{W}_{i,:}^{\text{new}})$ and $(\sigma_{:,j}^{\text{new}})^2 = \text{Var}(\mathcal{W}_{:,j}^{\text{new}})$ may continue to change. The increase in variance can make it more challenging to sample the weight parameters, that can meet the task objective (loss), from $\mathcal{W}_{i,:}^{\text{new}}$ and $\mathcal{W}_{:,j}^{\text{new}}$ due to their increased variability. To understand the behavior of $\text{Var}(\mathcal{W}_{i,:}^{\text{new}})$, we first examine the correlation $\rho_{i,:}$ between $\mathcal{W}_{i,:}$ and $-\eta \nabla \mathcal{W}_{i,:}$ in Eq. (6). It determines whether the newly obtained variance $\text{Var}(\mathcal{W}_{i,:}^{\text{new}})$ is greater than the previous $\text{Var}(\mathcal{W}_{i,:})$, depending on the following condition:

$$\eta^2 \text{Var}(\nabla \mathcal{W}_{i,:}) + 2\rho_{i,:}\text{Std}(\mathcal{W}_{i,:})|\eta|\text{Std}(\nabla \mathcal{W}_{i,:}) > 0. \tag{7}$$

From this, it can be seen that $\text{Var}(\mathcal{W}_{i,:}^{\text{new}}) > \text{Var}(\mathcal{W}_{i,:})$ if $\rho_{i,:} > |\eta|\text{Std}(\nabla\mathcal{W}_{i,:}) \; / \; 2\text{Std}(\mathcal{W}_{i,:})$. Likewise, $\text{Var}(\mathcal{W}_{:,j}^{\text{new}}) > \text{Var}(\mathcal{W}_{:,j})$ when the $\rho_{:,j}$ is also under $\rho_{:,j} > |\eta|\text{Std}(\nabla\mathcal{W}_{:,j})/2\text{Std}(\mathcal{W}_{:,j})$.

Thus, the range of $\text{Var}(\mathcal{W}_{i,:})$ and $\text{Var}(\mathcal{W}_{:,j})$ may keep changing (either increasing or decreasing), depending on the weight configuration and dataset, potentially affecting learning performance. However, owing to the similarities between GDU's distribution learning process and the gradient descent, GDU is likely to achieve competitive performance.

## 3.2 Weight Distribution Update (WDU)

Unlike GDU that learns the Isotropic Gaussian distribution of the weight gradient, Weight Distribution Update (WDU) learns the *distribution of weight parameters* after gradients have been applied to each weight parameter. Similar to GDU, $\mathbf{W}_{i,j}$ is first randomly sampled from $\mathcal{W}_{i,j} \sim \mathcal{N}(\mu_{i,j}, \sigma_{i,j}^2)$ generated using $\mathcal{W}_{i,:}$ and $\mathcal{W}_{:,j}$. Then, the $i$th row of the updated weight $\mathbf{W}_{i,:}^{\text{new}}$ is obtained from Eq. (3), followed by computing its mean and variance as $\mathcal{W}_{i,:}^{\text{new}} \sim \mathcal{N}(\mu_{i,:}^{\text{new}}, (\sigma_{i,:}^{\text{new}})^2)$. In the same way, the $j$th column of the updated weight $\mathbf{W}_{:,j}^{\text{new}}$ is obtained by computing its mean and variance as $\mathcal{W}_{:,j}^{\text{new}} \sim \mathcal{N}(\mu_{:,j}^{\text{new}}, (\sigma_{:,j}^{\text{new}})^2)$. Algorithm 2 in Section C shows the procedure of WDU.

Since WDU first samples random variables from a Gaussian distribution and then finds the distribution again, the central limit theorem (Fischer, 2011) applies during its process. That is, with WDU, the distribution $\mathcal{W}_{i,:} \sim \mathcal{N}(\mu_{i,:}, \sigma_{i,:}^2)$ is updated to $\mathcal{W}_{i,:}^{\text{new}} \sim \mathcal{N}(\mu_{i,:}^{\text{new}}, (\sigma_{i,:}^{\text{new}})^2) = \mathcal{N}(\mu_{i,:}, \frac{\sigma_{i,:}^2}{m})$, where $m$ is the sampling size. If sampling is repeated $k$ times, the $k$th distribution becomes $\mathcal{W}_{i,:}^{\text{new}} \sim \mathcal{N}(\mu_{i,:}^{\text{new}}, (\sigma_{i,:}^{\text{new}})^2) = \mathcal{N}(\mu_{i,:}, \frac{\sigma_{i,:}^2}{km})$, during which the variance decreases. This applies equally to $\mathcal{W}_{:,j}^{\text{new}}$, which decreases as the update is repeated. Thus, the variances $\sigma_{i,:}^2$ and $\sigma_{:,j}^2$ obtained with WDU is expected to be relatively smaller compared to GDU, implying that the behavior of the corresponding fully connected layer exhibits reduced variability.

The simultaneous use of row- and column-wise Isotropic distributions yields a unique characteristic in WDU. When constructing the distribution of each weight parameter, the row- and column-wise distributions are first combined as in Eq. (2). During the standard gradient descent process, the row-wise mean is drawn toward the column-wise mean, while the column-wise mean is attracted toward the row-wise mean. This mutual interaction creates an averaging loop between the two row- and column-wise perspectives. Formally, the update of $\mu_{i,:}$ and $\mu_{:,j}$ can be expressed as:

$$\mu_{i,:}^{(t+1)} = \frac{m}{m+n}\mu_{i,:}^{(t)} + \frac{n}{m+n}\bar{\mu}_{\text{col}}^{(t)} \quad \text{where} \quad \bar{\mu}_{\text{col}}^{(t)} = \frac{1}{m}\sum_j \mu_{:,j}^{(t)} \tag{8}$$

$$\mu_{:,j}^{(t+1)} = \frac{m}{m+n}\bar{\mu}_{\text{row}}^{(t)} + \frac{n}{m+n}\mu_{:,j}^{(t)} \quad \text{where} \quad \bar{\mu}_{\text{row}}^{(t)} = \frac{1}{n}\sum_i \mu_{i,:}^{(t)} \tag{9}$$

where $\mu_{i,:}^{(t)}$ and $\mu_{i,:}^{(t+1)}$ denote the row-wise mean at iteration $t$ and $t+1$, and $\mu_{:,j}^{(t)}$ and $\mu_{:,j}^{(t+1)}$ for the column-wise mean. The update rule encourages the alignment of row- and column-wise means; each row mean is influenced by corresponding column means, and conversely, and each column mean is drawn toward the row means. This mutual interaction ensures the distributions of rows and columns become progressively more similar over iterations. In contrast, in GDU, the row- and column-wise means can evolve independently, as the distributions are updated directly from the gradient.

## 3.3 Comparison between GDU and WDU

Both GDU and WDU update distributions, but they differ in how they model the concept of change. GDU emphasizes 'process of change'; it directly estimates gradient distributions and incorporates them into the existing distribution, capturing uncertainty. In contrast, WDU focuses on 'result of change'; it samples parameters from the current distribution, applies the gradient update, and then obtains a new distribution from the updated parameters. This distinction leads to different behaviors. WDU's variance generally decreases over time, promoting learning stability. GDU's variance, however, tends to fluctuate, often increasing, and remains sensitive to data and model properties, though standardization can mitigate this effect (see Section E). Thus, GDU encourages dynamic and exploratory learning that can potentially discover better optima but with reduced stability, whereas WDU favors predictable and stable learning, sometimes at the cost of overly conservative solutions.

## 4 EXPERIMENT

We experiment with the proposed Isotropic Gaussian learning on Vision Transformer (ViT) (Dosovitskiy et al., 2020), BERT, BERT-Large (Devlin et al., 2019), GPT-2 Small (Radford et al., 2019), MobileNetV2 (Sandler et al., 2018), ResNet-18 (He et al., 2016), and an 8-layer fully connected networks. We evaluate them with four tasks: 1) image classification, i.e., SVHN (Netzer et al., 2011), Food-101 (Bossard et al., 2014), mini-ImageNet (Vinyals et al., 2016), CIFAR-10 (Krizhevsky et al., 2009), and MNIST (Deng, 2012), 2) text classification, i.e., AG News (Zhang et al., 2015), Yelp Review (Zhang et al., 2015), and Emotions (Saravia et al., 2018), 3) text summarization, i.e., SAMSum (Gliwa et al., 2019), and 4) Question Answering, i.e., SQuAD v1.1 (Rajpurkar et al., 2016). We use a total of three GPUs, i.e., NVIDIA RTX 3090, RTX 4090, and A6000. Due to the inherent variability introduced by the sampling-based approach using distributions, experimental results may fluctuate under different settings. To ensure robustness and reliability, each experiment is conducted using at least ten different seeds, and the final reported results represent the average across these runs. The detailed experimental setups and comprehensive results are given in Section G and H, respectively. The code is provided at an anonymous GitHub repository in Section F.

### 4.1 TRANSFORMERS

**Vision Transformers (ViTs).** We evaluate Isotopic Gaussian learning on ViTs (Dosovitskiy et al., 2020), consisting of a total of twelve encoders, each containing six fully connected operations (Figure 2), i.e., query, key, and value in the attention module, as well as the self-output, intermediate, and output layers in the feed-forward module. We apply Isotropic Gaussian learning to all six operations. As shown in Table 1, on mini-ImageNet (Vinyals et al., 2016), accuracy is 76.77% with one encoder, 73.72% with two, and 70.78% with three, compared to 78.46% for the baseline. It reduces weight parameters in a single encoder from 7,087,872 to 38,032 (0.54%). When extended across multiple encoders, the proposed WDU and GDU preserve performance to a considerable degree, indicating their potential for application in complex Transformer architectures.

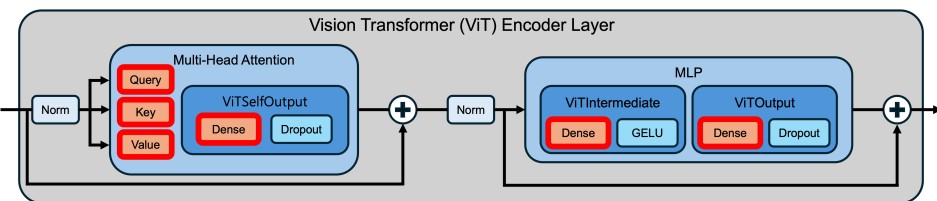

Figure 2: The structure of Vision Transformer (ViT) encoders. The Isotropic Gaussian learning is applied to the components highlighted in red within the orange-marked modules.

Table 1: The performance of ViT (Dosovitskiy et al., 2020) fine-tuned on mini-ImageNet (Vinyals et al., 2016) and Food-101 (Bossard et al., 2014). The 'base' refers to the model trained using backpropagation, and 'encoder layers' indicates the encoder with Isotropic Gaussian learning applied.

| encoder layers | base | 4 | 5 | 6 | 5,12 | 6,12 | 7,12 | 4,8,12 |
|---|---|---|---|---|---|---|---|---|
| mini-ImageNet (Vinyals et al., 2016) (%) | 78.46 | 76.77 | 76.14 | 75.95 | 73.09 | 73.72 | 73.04 | 70.78 |
| Food-101 (Bossard et al., 2014) (%) | 84.67 | 83.98 | 83.66 | 83.87 | 82.33 | 82.65 | 82.32 | 80.44 |
| Parameter (weight parameters vs. Isotropic mean and variance) | 7.09M per encoder layer | 38.03K × 1 | 38.03K × 1 | 38.03K × 1 | 38.03K × 2 | 38.03K × 2 | 38.03K × 2 | 38.03K × 3 |

**Language models.** We extend Isotropic Gaussians to language models including BERT, BERT-Large (Devlin et al., 2019), and GPT-2 Small (Radford et al., 2019). Similar to ViT, BERT comprises twelve encoders, each containing six feed-forward operations, including attention-related linear transformations and fully connected modules. We apply Isotropic Gaussian learning to BERT in the same manner as in the ViT experiment, targeting all six linear components within each encoder, allowing us to assess the generalizability of the Isotropic Gaussian across different types of models and tasks. As shown in Table 2, model accuracy remains competitive on the Emotions dataset: 94.16% with one encoder, 94.10% with three, and 93.99% with six, compared to 94.29% for the baseline. Moreover, applying Isotropic Gaussian learning to a single encoder in BERT reduces weight parameters from 7.09M to 38.03K (0.54% of baseline).

GPT-2 Small (Radford et al., 2019) employs a decoder-only architecture with twelve decoders, each comprising four linear matrices, including those used in the attention mechanism. We apply

Table 2: The performance of BERT (Devlin et al., 2019) fine-tuned on Emotions (Saravia et al., 2018), AG News (Zhang et al., 2015), and Yelp Review (Zhang et al., 2015). The 'base' denotes the models trained with back-propagation, and 'encoder layers' indicates the encoders with the Isotropic Gaussian applied.

| encoder layers | base | 6 | 9 | 5,12 | 8,12 | 4,6,8 | 5,7,9 | 4,8,10,12 | 2,4,6,8,10 | 2,4,6,8,10,12 |
|---|---|---|---|---|---|---|---|---|---|---|
| Emotions (Saravia et al., 2018) (%) | 94.29 | 94.13 | 94.16 | 94.11 | 94.12 | 94.10 | 94.02 | 94.08 | 94.04 | 93.99 |
| AG News (Zhang et al., 2015) (%) | 94.79 | 94.52 | 94.27 | 94.30 | 94.31 | 94.16 | 94.22 | 94.14 | 93.98 | 93.69 |
| Yelp Review (Zhang et al., 2015) (%) | 88.78 | 88.21 | 88.18 | 87.98 | 87.80 | 87.55 | 87.41 | 87.30 | 86.82 | 86.94 |
| Parameter (weight parameters vs. Isotropic mean and variance) | 7.09M per encoder layer | 38.03K × 1 | 38.03K × 1 | 38.03K × 2 | 38.03K × 2 | 38.03K × 3 | 38.03K × 3 | 38.03K × 4 | 38.03K × 5 | 38.03K × 6 |

Table 3: The performance of GPT-2 Small (Radford et al., 2019) fine-tuned on SAMSum dataset (Gliwa et al., 2019) . The 'base' refers to the model trained using back-propagation, and 'decoder layers' (bold numbers) indicate the decoders with Isotropic Gaussian learning applied.

| decoder layers | base | 6 | 10 | 12 | 8,12 | 9,12 | 10,12 | 6,8,10 | 4,6,8,10 |
|---|---|---|---|---|---|---|---|---|---|
| BLEU (Papineni et al., 2002) | 0.6957 | 0.6724 | 0.6969 | 0.6963 | 0.6902 | 0.6941 | 0.6935 | 0.6509 | 0.6685 |
| ROUGE-1 (Lin, 2004) | 0.7466 | 0.7116 | 0.7315 | 0.7404 | 0.7274 | 0.7321 | 0.7287 | 0.6701 | 0.6653 |
| ROUGE-L (Lin, 2004) | 0.7419 | 0.7066 | 0.7246 | 0.7327 | 0.7218 | 0.7269 | 0.7226 | 0.6637 | 0.6603 |
| Parameter (weight parameters vs. Isotropic mean and variance) | 7.09M per decoder layer | 37.63K × 1 | 37.63K × 1 | 37.63K × 1 | 37.63K × 2 | 37.63K × 2 | 37.63K × 2 | 37.63K × 3 | 37.63K × 4 |

Table 4: The performance of BERT-Large (Devlin et al., 2019) fine-tuned on SQuAD v1.1 (Rajpurkar et al., 2016), comparing the back-propagation baseline (base) with models where Isotropic Gaussian learning is applied to different encoder layers. The 'bold numbers' mean the number of encoder layers to which Isotropic Gaussian learning is applied (see Section H for details).

| encoder layers | base | 1 | 2 | 3 | 4 | 5 | 6 | 7 | 8 |
|---|---|---|---|---|---|---|---|---|---|
| Exact Match (EM) | 84.42 | 82.55 | 82.24 | 82.96 | 80.25 | 74.83 | 79.06 | 73.32 | 75.57 |
| F1 | 90.93 | 89.72 | 89.22 | 89.79 | 87.54 | 83.58 | 87.04 | 82.31 | 83.78 |
| Parameter (weight parameters vs. Isotropic mean and variance) | 12.06M per encoder layer | 50.18K × 1 | 50.18K × 2 | 50.18K × 3 | 50.18K × 4 | 50.18K × 5 | 50.18K × 6 | 50.18K × 7 | 50.18K × 8 |

Isotropic Gaussian learning to all of these matrices, reducing the parameters per decoder from 7.09M to 37.63K (0.53% of the baseline). Experiments are conducted on the SAMSum dataset (Gliwa et al., 2019), with model performance evaluated using ROUGE (Lin, 2004) and BLEU (Papineni et al., 2002). As shown in Table 3, performance remains nearly identical to the baseline when one decoder is modified and shows only minor degradation with two or three, with more noticeable drops occurring beyond three layers. These results indicate the potential of Isotropic Gaussian learning to be applied effectively not only to classification but also to more complex tasks such as summarization.

BERT-Large (Devlin et al., 2019) differs from BERT in that it contains 24 encoder layers and larger feed-forward operations. Applying Isotropic Gaussian learning to BERT-Large reduces the number of parameters per encoder from 12.06M to 50.18K (0.40%). Table 4 presents the experimental results obtained using BERT-Large on the SQuAD v1.1 (Rajpurkar et al., 2016) dataset. With three encoder layers applied, Exact Match (EM) decreased slightly from 84.42 to 82.96, and the F1 score from 90.93 to 89.79. However, extending Isotropic Gaussian learning to additional fully connected layers leads to a gradual decline in model performance, though overall stability is maintained.

## 4.2 CONVOLUTIONAL NEURAL NETWORKS

Convolution filters are weight groups that detect patterns in images, but their sliding nature limits GPU parallelism. The im2col (Vasudevan et al., 2017) unfolds image patches into vectors and reshaping filters into matrices, turning convolution into efficient matrix multiplication. We apply Isotropic Gaussian learning to the im2col-transformed matrix representation in our experiments. Table 5 reports accuracy of MobileNetV2 (Sandler et al., 2018) and ResNet-18 (He et al., 2016) on CIFAR-10 (Krizhevsky et al., 2009) and SVHN (Netzer et al., 2011). Applying the method to a single convolution module reduces weight parameters to 0.78%–22.25% in MobileNetV2 and 0.43%–4.69% in ResNet-18. With fewer than four modules, model performance remains close to the baseline, while larger applications leads to degradation, with detailed results in Section H.2.

Both MobileNetV2 and ResNet-18 consist of convolutional backbones, followed by a pooling layer and a classifier that often includes a fully connected layer. We modify the classifier by replacing it with two fully connected layers and applying Isotropic Gaussian learning to the first layer. This reduces the number of classifier weight parameters to depend only on the hidden size $h$, specifically

Table 5: The performance of MobileNetV2 (Sandler et al., 2018) and ResNet-18 (He et al., 2016) fine-tuned on CIFAR-10 (Krizhevsky et al., 2009) and SVHN (Netzer et al., 2011). The 'base' refers to the model trained via back-propagation, and the 'bold numbers' indicate the number of convolution modules/layers to which Isotropic Gaussian learning is applied. The specific combinations of modules and layers to which the methodology is applied are detailed in the Section H.

| Model | Data | base | 1 | 3 | 4 | 5 | 6 | 8 |
|---|---|---|---|---|---|---|---|---|
| MobileNetV2 (Sandler et al., 2018) | CIFAR-10 (%) (Krizhevsky et al., 2009) | 94.94 | 94.81 | 91.20 | 90.99 | 93.67 | 87.28 | 86.81 |
| | SVHN (%) (Netzer et al., 2011) | 96.58 | 96.54 | 94.92 | 95.63 | 94.25 | 94.29 | 92.71 |
| ResNet-18 (He et al., 2016) | CIFAR-10 (%) (Krizhevsky et al., 2009) | 94.64 | 94.85 | 93.41 | 90.22 | 89.89 | 89.43 | 82.19 |
| | SVHN (%) (Netzer et al., 2011) | 96.57 | 96.45 | 96.34 | 96.07 | 90.59 | 91.10 | 88.92 |

Table 6: The performance of MobileNetV2 (Sandler et al., 2018) and ResNet-18 (He et al., 2016) fine-tuned on CIFAR-10 (Krizhevsky et al., 2009) and SVHN (Netzer et al., 2011). The 'base' refers to the model trained via back-propagation, and 'Isotropic' denotes the model trained with an Isotropic Gaussian. 'number of parameters' refers to those in the classifier layer, i.e., weight parameters and Isotropic means and variances for each. The hidden length $h$ is 32.

| Model | Method | CIFAR-10 (Krizhevsky et al., 2009) | | SVHN (Netzer et al., 2011) | | Number of |
|---|---|---|---|---|---|---|
| | | Train Acc (%) | Test Acc (%) | Train Acc (%) | Test Acc (%) | parameters |
| MobileNetV2 (Sandler et al., 2018) | base | 99.28 | 94.87 | 99.70 | 96.42 | 12.81K |
| | Isotropic | 97.09 | 91.68 | 95.52 | 93.73 | 2986 |
| ResNet-18 (He et al., 2016) | base | 99.17 | 94.47 | 99.60 | 96.57 | 5.13K |
| | Isotropic | 98.46 | 90.23 | 98.88 | 94.04 | 1450 |

$13h + 2,570$ for CIFAR-10 and SVHN. As shown in Table 6, MobileNetV2 uses only 23.31% of the original classifier weight parameters while accuracy decreases slightly from 94.87% to 91.68% on CIFAR-10 and from 96.42% to 93.73% on SVHN. For ResNet-18, the model accuracy reaches 90.23% on CIFAR-10 and 94.04% on SVHN with just 1,450 parameters (28.87% of baseline).

## 4.3 FULLY CONNECTED NETWORKS

To understand the properties and behaviors of the proposed Isotropic Gaussian learning more deeply, we apply it to a network solely consisting of eight fully connected layers on MNIST. As shown in Table 7, applying it to four layers reduces the number of parameters from 16,384 to 512 per layer (3.13% of baseline) with only a slight drop in accuracy (from 95.81% to 95.78%). However, when applied to either the first or the last layer, larger performance losses are observed (Table 23 in Section H), reflecting their critical roles in feature extraction and classification. These results suggest that careful layer selection is necessary, as excessive application of Isotropic Gaussians may lead to overfitting and impair the network's generalization ability, resulting in relatively lower test accuracy compared to training accuracy. This limitation could potentially be mitigated through regularization, Gaussian blur (Stockman & Shapiro, 2001), or dropout (Srivastava et al., 2014).

Table 7: The train and test accuracy of 8-layer fully networks on MNIST (Deng, 2012) consisting of 128 neurons per each layer, with the number of weight parameters and Isotropic Gaussian parameters (mean and variance), and their ratio for each layer. Here, 'base' refers to back-propagation training, and 'layers' (bold numbers) indicates the layers applying Isotropic Gaussian learning.

| layers | base | 5 | 6 | 7 | 3,6 | 4,6 | 4,7 | 3,5,7 | 3,4,5,6 | 4,5,6,7 |
|---|---|---|---|---|---|---|---|---|---|---|
| train acc(%) | 99.94 | 96.46 | 94.89 | 97.14 | 96.72 | 96.58 | 98.75 | 96.85 | 93.49 | 96.76 |
| test acc(%) | 95.81 | 95.62 | 94.84 | 96.76 | 94.87 | 95.60 | 96.43 | 93.53 | 93.15 | 95.78 |
| Parameter (weight parameters vs. Isotropic mean and variance) | 16.4k per layer | $512 \times 1$ | $512 \times 1$ | $512 \times 1$ | $512 \times 2$ | $512 \times 2$ | $512 \times 2$ | $512 \times 3$ | $512 \times 4$ | $512 \times 4$ |

## 4.4 COMPARISON TO ALTERNATIVE DISTRIBUTIONS

To assess the necessity and effectiveness of learning Isotropic Gaussian distributions, we compare WDU and GDU with three alternative distributions: (1) weight parameters sampled from backpropagation-based distributions, (2) random Gaussian weight parameters, and (3) models with skipped layers. The mean and standard deviation are randomly chosen from [-1, 1] and [1e-12, 1], respectively, and the results are averaged over 1,000 runs. As in Table 8, both WDU and GDU retain over 95.78% accuracy when applied to up to four layers. In contrast, the backpropagation-based distribution quickly collapses to about 12% accuracy, and random sampling hovers near 10%. Models with skipped layers perform slightly better than them but reaches only 21.40% at best. The results indicate that WDU and GDU effectively model the underlying Isotropic Gaussian distributions of fully connected layers, while alternative distributions provide only limited representational capacity.

Table 8: The performance when weight parameters are sampled from distributions obtained via back-propagation (BP), random distributions, and layer removal, compared with WDU and GDU. Here, the bold numbers indicate which layers each method is applied to.

| layers | 6 | 5,7 | 3,5,7 | 5,6,7 | 4,5,6,7 | 3,4,5,6,7 |
|---|---|---|---|---|---|---|
| WDU/GDU (%) | 94.84 | 97.39 | 93.53 | 87.56 | 95.78 | 85.37 |
| BP (%) | 12.32 | 12.60 | 12.11 | 12.49 | 12.18 | 11.66 |
| random (%) | 11.09 | 10.49 | 10.50 | 10.51 | 10.55 | 10.57 |
| remove (%) | 11.35 | 11.35 | 11.35 | 21.20 | 21.40 | 19.30 |

## 5 RELATED WORK

**Bayesian neural networks.** The proposed Isotropic Gaussian learning is analogous to Bayesian neural networks (BNNs) (Kononenko, 1989; Charnock et al., 2022) in that it estimates weight distributions and leverages them for sampling during model execution. Bayes by Backprop (Blundell et al., 2015), a representative method based on BNNs, approximates the posterior distribution of weights through variational inference. While BNNs often adopt Isotropic Gaussian distributions (Graves, 2011), these are typically employed only as prior distributions; after learning, they generally transform into different distributional forms, increasing the parameter count from $n^2$ to $2n^2$. In contrast, the proposed approach retains the Isotropic Gaussian representation after learning and reduces the number of parameters to $4n$, asymptotically achieving $\mathcal{O}(n)$ memory efficiency.

**Model compression.** The method proposed in this paper reduces parameters by exploiting weight distributions, offering effects similar to compression methods (Cheng et al., 2017) such as quantization (Hubara et al., 2018), distillation (Hinton et al., 2015), and pruning (Han et al., 2015). Unlike them using fixed parameters, it adapts flexibly to data and models. For example, applied to the ViT (Dosovitskiy et al., 2020) encoder, it achieves comparable performance with only 0.54% of the original parameters and just a 0.69% accuracy drop, demonstrating its efficiency and effectiveness.

**Randomized neural networks.** Random Vector Functional Links (RVFLs) (Gallicchio et al., 2018) add randomness through fixed random feature layers (Scardapane & Wang, 2017), but they require storing fixed weights and train only the readout layer. In contrast, our method stores only the mean and variance of an Isotropic distribution, samples weights at each execution, and trains all layers, which adapts to the distribution via repeated sampling, unlike RVFLs. Similarly, structured random features and rainbow networks (Guth et al., 2024) rely on random feature maps and covariance multiplications, while our approach directly generates parameters from learned Isotropic distributions.

**Structural pruning.** Structural pruning (Sun et al., 2025; Zhang et al., 2024; Gromov et al., 2024; He et al., 2024; Liao et al., 2025; Pons et al., 2024) reduces model computation by removing redundant layers, mainly in Transformers. It modifies the model architecture to answer "Which parts of the model are unnecessary?" In contrast, the proposed method replaces weight parameters of fully connected layers with simplified Isotropic Gaussians rather than deleting layers, which can be applied to any model employing fully connected operations. In particular, this study tries to answer the question "Do weights need to be stored individually?" and investigates the statistical perspective of fully connected operations, leading to both theoretical insights and significant memory savings.

## 6 LIMITATIONS AND DISCUSSIONS

Due to page constraints, we provide the limitations and discussions of this work in Section B.

## 7 CONCLUSION

We present a new statistical perspective on fully connected layers of neural networks by modeling them with Isotropic Gaussian distributions. This perspective not only provides fresh insights into the statistical properties of fully connected layers but also enables highly efficient compression of their weight parameters. We introduce two Isotropic Gaussian learning methods, GDU and WDU, which demonstrate that a weight matrix can be effectively represented using row- and column-wise Isotropic Gaussians. This representation reduces the parameter complexity from $\mathcal{O}(n^2)$ to $\mathcal{O}(n)$ with minimal impact on model performance when used selectively. The seamless integration of the proposed method with standard backpropagation also underscores its practical applicability.

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

## A    The Use of Large Language Models (LLMs)

We used large language models solely for polishing grammar and improving the readability of the manuscript. The contents and research contributions were written entirely by the authors.

## B    Limitations and discussions

**Limitations of distribution-based representation.** This work demonstrates the potential of representing fully connected layers with an Isotropic Gaussian distribution, reducing memory requirements from $\mathcal{O}(n^2)$ to $\mathcal{O}(n)$ and achieving up to 251.04× savings in the parts where our method is applied, as demonstrated in our Transformer experiments. However, an Isotropic Gaussian distribution cannot fully capture the expressive capacity of a weight matrix. Even with the application of both row-wise and column-wise perspectives, the experimental results show that representing the entire model solely with Isotropic Gaussian distributions remains challenging. This points to inherent upper bounds of the distributional approach, which should be examined in more detail. Therefore, further research is required to determine the feasibility of using minimal-parameter distributions in place of full parameter sets.

**Performance improvement.** Through experiments, we show that modeling each row and column of the weight matrix as an independent Isotropic Gaussian enables effective learning with fewer parameters. In particular, this approach demonstrated high efficiency in specific subsets of fully connected layers. However, the Isotropic Gaussian distribution is insufficient to capture complex correlation structures, leading to performance degradation in more challenging tasks. To address this issue, future work should explore richer distributions or covariance structures, while balancing improved expressiveness against the added parameter cost.

## C    GDU and WDU learning algorithm

Algorithm 1 describes Gradient Distribution Update (GDU), while Algorithm 2 outlines Weight Distribution Update (WDU). Let the size of all weights matrix be $[n, m]$ ($n$: number of row, $m$: number of column), the total number of data be $N$, and the total number of network parameters be $P$. Then, the time complexity of GDU and WDU becomes $\mathcal{O}(tnm) + N[\mathcal{O}(tnm) + \mathcal{O}(P) + \mathcal{O}(P - tnm)]$ and $\mathcal{O}(tnm) + N[\mathcal{O}(tnm) + \mathcal{O}(P)]$, respectively. As the model size $P$ and dataset size $N$ grow, both GDU and WDU asymptotically require up to twice the computational time of the gradient descent method, which has a time complexity of $N \times \mathcal{O}(P)$.

## D    Standardization

To promote stable distribution learning in both forward and backward propagation, the standardization of the input (Mendenhall et al., 2016) can be considered, which normalizes the mean and variance of the layer input distribution on a row-wise basis to zero and one, respectively.

**Forward propagation.** For a fully connected layer, expressed as $y_i = \mathbf{W}_{i,:}\mathbf{x} + b_i$ for $i \in \{1, 2, \ldots, m\}$, the Isotropic Gaussian distribution of the output $y_i$, i.e., the mean $\mu_{y_i}$ and variance $\sigma^2_{y_i}$, is given by:

$$\mu_{y_i} = \mathbb{E}[y_i] = \mathbb{E}[\mathbf{W}_{i,:}\mathbf{x} + b_i] = \mathbb{E}[\mathbf{W}_{i,:}\mathbf{x}] + \mathbb{E}[b_i] = \mathbb{E}[\sum_{j=1}^{n} w_{i,j}x_j] + \mathbb{E}[b_i] = n\mathbb{E}[\mathbf{W}]\mathbb{E}[\mathbf{x}] + b_i$$

$$(10)$$

$$\sigma^2_{y_i} = \mathrm{Var}(y_i) = \mathrm{Var}(\mathbf{W}_{i,:}\mathbf{x} + b_i) = \mathrm{Var}(\mathbf{W}_{i,:}\mathbf{x}) = \mathrm{Var}(\sum_{j=1}^{n} w_{i,j}x_j) = \sum_{j=1}^{n} \mathrm{Var}(w_{i,j}x_j)$$

$$= n[\mathrm{Var}(\mathbf{W})\mathbb{E}[\mathbf{x}]^2 + \mathbb{E}[\mathbf{W}]^2\mathrm{Var}(\mathbf{x}) + \mathrm{Var}(\mathbf{W})\mathrm{Var}(\mathbf{x})]$$

$$(11)$$

---

**Algorithm 1** Gradient distribution update (GDU)

---

**Input:** target weight $\mathcal{T} = \{\mathbf{W}^{(1)}, \mathbf{W}^{(2)}, \ldots, \mathbf{W}^{(t)}\}$, learning rate $\eta$, dataset $(\mathbf{X}, \mathbf{Y})$, Objective function $\mathcal{L}$
**Output:** Final mean $\mu = \{\mu^{(1)}, \mu^{(2)}, \ldots, \mu^{(t)}\}$ and updated std $\sigma = \{\sigma^{(1)}, \sigma^{(2)}, \ldots, \sigma^{(t)}\}$
**procedure** INITIAL STATE
    **for** $i \in \{1, 2, \ldots, t\}$ **do**
        $n^{(i)}, m^{(i)}$ = number of row and column in $\mathcal{W}^{(i)}$
        **for** $j \in \{1, 2, \ldots, n^{(i)}\}$ **do**
            $\mu_{j,:}^{(i)}, \sigma_{j,:}^{(i)} = \text{Mean}(\mathcal{W}_{j,:}^{(i)}), \text{Std}(\mathcal{W}_{j,:}^{(i)})$
        **end for**
        **for** $k \in \{1, 2, \ldots, m^{(i)}\}$ **do**
            $\mu_{:,k}^{(i)}, \sigma_{:,k}^{(i)} = \text{Mean}(\mathcal{W}_{:,k}^{(i)}), \text{Std}(\mathcal{W}_{:,k}^{(i)})$
        **end for**
    **end for**
**end procedure**
**for** $(\mathbf{x}, \mathbf{y}) \in (\mathbf{X}, \mathbf{Y})$ **do**
    **procedure** FORWARD-PROPAGATION
        **for** $i \in \{1, 2, \ldots, t\}$ **do**
        $n^{(i)}, m^{(i)}$ = number of row and column in $\mathcal{W}^{(i)}$
        **for** $j \in \{1, 2, \ldots, n^{(i)}\}$ **do**
            **for** $k \in \{1, 2, \ldots, m^{(i)}\}$ **do**
                $\mu_{j,k}^{(i)} = \frac{m^{(i)}\mu_{j,:} + n^{(i)}\mu_{:,k}}{m^{(i)} + n^{(i)}}$
                $(\sigma_{j,k}^{(i)})^2 = \frac{1}{m^{(i)}+n^{(i)}}(((m^{(i)}-1)\sigma_{j,:}^2 + (n^{(i)}-1)\sigma_{:,k}^2) + \frac{m^{(i)}n^{(i)}}{(m^{(i)}+n^{(i)})^2}(\mu_{j,:} - \mu_{:,k})^2$
                $\mathcal{W}_{j,k}^{(i)} = \textbf{Initialize}(\mathcal{W}_{j,k}^{(i)}, \mu_{j,k}^{(i)}, \sigma_{j,k}^{(i)})$
            **end for**
        **end for**
        **end for**
        Compute the predicted output $\hat{\mathbf{y}}$
        Compute the loss $L = \mathcal{L}(\hat{\mathbf{y}}, \mathbf{y})$
    **end procedure**
    **procedure** BACK-PROPAGATION
        Compute the gradient of all parameters
        **for** $i \in \{1, 2, \ldots, t\}$ **do**
        $n^{(i)}, m^{(i)}$ = number of row and column in $\mathcal{W}^{(i)}$
        **for** $j \in \{1, 2, \ldots, n^{(i)}\}$ **do**
            $\rho_{j,:} = \text{corr}(\mathcal{W}_{j,:}^{(i)}, -\eta\nabla\mathcal{W}_{j,:}^{(i)})$
            $\Sigma = \text{Std}(-\eta\nabla\mathcal{W}_{j,:}^{(i)})$
            $\mu_{j,:}^{(i)} \leftarrow \mu_{j,:}^{(i)} + \text{Mean}(-\eta\nabla\mathcal{W}_{j,:}^{(i)})$
            $\sigma_{j,:}^{(i)} \leftarrow [(\sigma_{j,:}^{(i)})^2 + \Sigma^2 + 2\sigma_{j,:}^{(i)}\Sigma\rho_{j,:}]^{0.5}$
        **end for**
        **for** $k \in \{1, 2, \ldots, m^{(i)}\}$ **do**
            $\rho_{:,k} = \text{corr}(\mathcal{W}_{:,k}^{(i)}, -\eta\nabla\mathcal{W}_{:,k}^{(i)})$
            $\Sigma = \text{Std}(-\eta\nabla\mathcal{W}_{:,k}^{(i)})$
            $\mu_{:,k}^{(i)} \leftarrow \mu_{:,k}^{(i)} + \text{Mean}(-\eta\nabla\mathcal{W}_{:,k}^{(i)})$
            $\sigma_{:,k}^{(i)} \leftarrow [(\sigma_{:,k}^{(i)})^2 + \Sigma^2 + 2\sigma_{:,k}^{(i)}\Sigma\rho_{:,k}]^{0.5}$
        **end for**
        **end for**
        Update all parameters expect target weights using Gradient Descent
    **end procedure**
**end for**

---

Since the input $\mathbf{x}$ comes to follow the standard Gaussian (i.e., $\mathbb{E}[\mathbf{x}] = 0$ and $\text{Var}(\mathbf{x}) = 1$) through standardization, the distribution of the output $\mathbf{y}_i$ becomes:

$$\mathbb{E}[y_i] = b_i, \ \text{Var}(y_i) = n(\mathbb{E}[\mathbf{W}]^2 + \text{Var}(\mathbf{W})) \tag{12}$$

Therefore, when applied to the forward propagation, standardization ensures that 1) $y_i$ becomes independent of the distribution of the preceding layer, i.e., $\mathbf{x}$, and 2) $\text{Var}(y_i)$ becomes smaller when $\text{Var}(\mathbf{x}) > 1$.

---

**Algorithm 2** Weight distribution update (WDU)

---

**Input:** target weight $\mathcal{T} = \{\mathbf{W}^{(1)}, \mathbf{W}^{(2)}, \ldots, \mathbf{W}^{(t)}\}$, learning rate $\eta$, data $(\mathbf{X}, \mathbf{Y})$, Objective function $\mathcal{L}$
**Output:** Final mean $\mu = \{\mu^{(1)}, \mu^{(2)}, \ldots, \mu^{(t)}\}$ and updated std $\sigma = \{\sigma^{(1)}, \sigma^{(2)}, \ldots, \sigma^{(t)}\}$
**procedure** INITIAL STATE
    **for** $i \in \{1, 2, \ldots, t\}$ **do**
        $n^{(i)}, m^{(i)}$ = number of row and column in $\mathcal{W}^{(i)}$
        **for** $j \in \{1, 2, \ldots, n^{(i)}\}$ **do**
            $\mu_{j,:}^{(i)}, \sigma_{j,:}^{(i)} = \text{Mean}(\mathcal{W}_{j,:}^{(i)}), \text{Std}(\mathcal{W}_{j,:}^{(i)})$
        **end for**
        **for** $k \in \{1, 2, \ldots, m^{(i)}\}$ **do**
            $\mu_{:,k}^{(i)}, \sigma_{:,k}^{(i)} = \text{Mean}(\mathcal{W}_{:,k}^{(i)}), \text{Std}(\mathcal{W}_{:,k}^{(i)})$
        **end for**
    **end for**
**end procedure**
**for** $(\mathbf{x}, \mathbf{y}) \in (\mathbf{X}, \mathbf{Y})$ **do**
    **procedure** FORWARD-PROPAGATION
        **for** $i \in \{1, 2, \ldots, t\}$ **do**
        $n^{(i)}, m^{(i)}$ = number of row and column in $\mathcal{W}^{(i)}$
        **for** $j \in \{1, 2, \ldots, n^{(i)}\}$ **do**
            **for** $k \in \{1, 2, \ldots, m^{(i)}\}$ **do**
                $\mu_{j,k}^{(i)} = \frac{m^{(i)}\mu_{j,:} + n^{(i)}\mu_{:,k}}{m^{(i)} + n^{(i)}}$
                $(\sigma_{j,k}^{(i)})^2 = \frac{1}{m^{(i)} + n^{(i)}}(((m^{(i)} - 1)\sigma_{j,:}^2 + (n^{(i)} - 1)\sigma_{:,k}^2) + \frac{m^{(i)}n^{(i)}}{(m^{(i)} + n^{(i)})^2}(\mu_{j,:} - \mu_{:,k})^2$
                $\mathcal{W}_{j,k}^{(i)} = \textbf{Initialize}(\mathcal{W}_{j,k}^{(i)}, \mu_{j,k}^{(i)}, \sigma_{j,k}^{(i)})$
            **end for**
        **end for**
        **end for**
        Compute the predicted output $\hat{\mathbf{y}}$
        Compute the loss $L = \mathcal{L}(\hat{\mathbf{y}}, \mathbf{y})$
    **end procedure**
    **procedure** BACKWARD-PROPAGATION
        Compute the gradient of all parameters
        Update all parameters
        **for** $i \in \{1, 2, \ldots, t\}$ **do**
        $n^{(i)}, m^{(i)}$ = number of row and column in $\mathcal{W}^{(i)}$
        **for** $j \in \{1, 2, \ldots, n^{(i)}\}$ **do**
            $\mu_{j,:}^{(i)}, \sigma_{j,:}^{(i)} = \text{Mean}(\mathcal{W}_{j,:}^{(i)}), \text{Std}(\mathcal{W}_{j,:}^{(i)})$
        **end for**
        **for** $k \in \{1, 2, \ldots, m^{(i)}\}$ **do**
            $\mu_{:,k}^{(i)}, \sigma_{:,k}^{(i)} = \text{Mean}(\mathcal{W}_{:,k}^{(i)}), \text{Std}(\mathcal{W}_{:,k}^{(i)})$
        **end for**
        **end for**
    **end procedure**
**end for**

---

**Backward propagation.** For a fully connected layer, expressed as $y_i = \mathbf{W}_{i,:}\mathbf{x} + b_i$, the gradient of the task objective (loss) $\mathcal{L}$ w.r.t. the weight $w_{i,j}$ and bias $b_i$ is given by:

$$\nabla w_{i,j} = \frac{\partial \mathcal{L}}{\partial w_{i,j}} = \frac{\partial \mathcal{L}}{\partial y_i}\frac{\partial y_i}{\partial w_{i,j}} = \nabla y_i x_j \tag{13}$$

$$\nabla b_i = \frac{\partial \mathcal{L}}{\partial b_i} = \frac{\partial \mathcal{L}}{\partial y_i}\frac{\partial y_i}{\partial b_i} = \nabla y_i \text{ where } \nabla y_i = \frac{\partial \mathcal{L}}{\partial y_i} \tag{14}$$

From Equation (14), it can be seen that the bias gradient $\nabla b_i$ only depends on subsequent layers that contribute to $\mathcal{L}$. In contrast, as shown in Equation (13), the weight gradient $\nabla w_{i,j}$ depends on both subsequent layers and the input $\mathbf{x}$, with the mean $\mu_{\nabla w_{i,j}}$ and variance $\sigma_{\nabla w_{i,j}}^2$, which is given by:

$$\mu_{\nabla w_{i,j}} = \mathbb{E}[\nabla w_{i,j}] = \mathbb{E}[\nabla y_i x_j] = \mathbb{E}[\nabla y_i]\mathbb{E}[x_j] \tag{15}$$

$$\sigma_{\nabla w_{i,j}}^2 = \text{Var}[\nabla w_{i,j}] = \mathbb{E}[(\nabla y_i x_j)^2] - (\mathbb{E}[\nabla y_i]\mathbb{E}[x_j])^2 = \mathbb{E}[\nabla y_i^2]\mathbb{E}[x_j^2] - \mathbb{E}[\nabla y_i]^2\mathbb{E}[x_j]^2 \tag{16}$$

With standardization, i.e., $\mathbb{E}[x_j] = 0$ and thus $\mathbb{E}[x_j^2] = \mathbb{E}[x_j]^2 + \text{Var}(x_j) = 1$, the mean and variance become:

$$\mathbb{E}[\nabla w_{i,j}] = 0, \ \text{Var}[\nabla w_{i,j}] = \mathbb{E}[\nabla y_i^2] \tag{17}$$

Consequently, when the input is standardized during the backward propagation, the weight gradient $\nabla w_{i,j}$ comes to depend only on the subsequent layers (i.e., $y_i$), likely leading to more stable and steady distribution learning.

## E  ISOTROPIC MEAN AND VARIANCE

**GDU and WDU comparison.** Figure 3 plots the changes in Isotropic mean and standard deviation (variance) learned by GDU and WDU over training epochs on CIFAR-10. As discussed in Section 3.1 and 3.2, WDU exhibits similar behavior across both row-wise and column-wise distributions, with the variance progressively approaching zero. In contrast, the GDU demonstrates independent updates for each row and column, and in most cases the variance tends to increase gradually over time. The relatively increasing variance observed in GDU can be partially stabilized through the application of standardization(Section D).

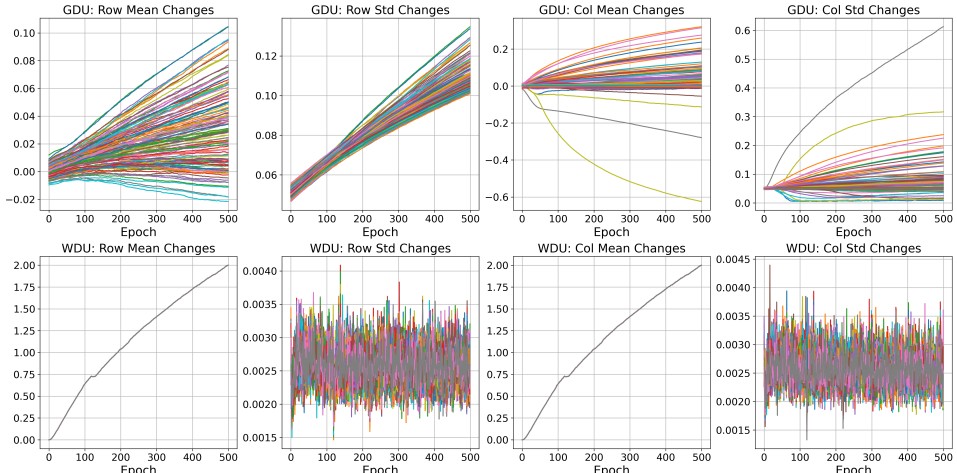

Figure 3: The change of Isotropic Gaussian mean and standard deviation (std) of GDU (above) and WDU (below) on CIFAR-10 (layer 3) over training epochs. In the graph, each line corresponds to a row and column of the weight matrix.

**Standardization.** Figure 4 shows the changes in Isotropic standard deviation (std) of the layer 5 in GDU over training epochs on CIFAR-10, with and without standardization (Section D). While row-wise standard deviation (std) decreased from 0.20 to 0.13 (about 1.5x), column-wise std decreased from 0.25 to 0.12 (about 2x), suggesting that standardization effectively stabilizes the Isotropic Gaussian distributions. This indicates that it effectively stabilizes the Isotropic Gaussian distributions. Considering that GDU may diverge with increasing mean and variance, the application of standardization can serve as an effective mechanism to mitigate divergence during Isotropic Gaussian learning.

## F  CODE IMPLEMENTATION

We provide the code implementation of the proposed Isotropic Gaussian learning and experimental details at an anonymous GitHub repository[2].

---

[2] https://anonymous.4open.science/r/isotropic_gaussian_learning

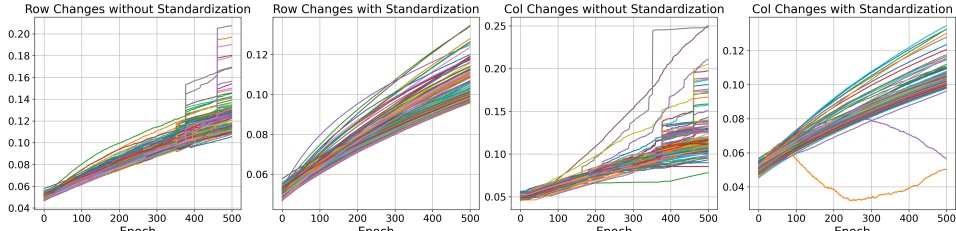

Figure 4: The changes in Isotropic mean and standard deviation (std) learned by GDU over training epoch on CIFAR-10 (layer 5). The first and second plots correspond to the row-wise distributions, while the third and fourth correspond to the column-wise distributions. In each case, the first and third plots illustrate results without standardization, whereas the second and fourth plots show results with standardization applied. In the graph, each line corresponds to a row/column of the weight matrix.

## G EXPERIMENTAL SETUPS

### G.1 DATASETS

In our experiment, we use five datasets for vision task, i.e. Food-101 (Bossard et al., 2014), mini-ImageNet (Vinyals et al., 2016), CIFAR-10 (Krizhevsky et al., 2009), Street View House Numbers (SVHN) (Netzer et al., 2011), and MNIST (Deng, 2012), and five datasets for language tasks, i.e. AG News (Zhang et al., 2015), Yelp Review (Zhang et al., 2015), Emotions (Saravia et al., 2018), SAMSum (Gliwa et al., 2019), and SQuAD v1.1 (Rajpurkar et al., 2016) to evaluate the performance of the proposed Isotropic Gaussian learning across different domains.

**Food-101.** Food-101 is a dataset comprising images from 101 distinct food categories. Each class contains 1,000 images, and the images vary in resolution.

**mini-ImageNet.** mini-ImageNet is a scaled-down version of the ImageNet dataset. In our experiments, we utilized 1,000 classes, with approximately 30 to 40 images per class.

**CIFAR-10.** CIFAR-10 is a dataset commonly used in image classification research. It contains 60,000 color images, each of size 32x32 pixels, divided into 10 distinct classes. The dataset is split into 50,000 training images and 10,000 testing images.

**SVHN.** SVHN is a dataset designed for recognizing digits (0 to 9) from house number images captured from a distance. Each image has a resolution of 32×32 pixels. The dataset contains approximately 600,000 images in total, including around 74,000 for training, 26,000 for testing, and the remainder designated as extra data.

**MNIST.** The MNIST dataset is a widely used benchmark for image classification tasks, consisting of 70,000 gray-scale images of handwritten digits, each of size 28x28 pixels. It is divided into 60,000 training images and 10,000 testing images, with 10 distinct classes corresponding to the digits 0 through 9.

**AG News.** AG News is a text classification dataset designed to categorize news articles into four distinct topics based on their titles and a portion of their content. Each class contains 30,000 training samples and 1,900 testing samples.

**Yelp Review.** The Yelp Review dataset is used for text classification, where user-written restaurant reviews are categorized based on their ratings from 1 to 5 stars. For our experiments, we grouped the ratings into three sentiment classes: 1–2 stars as negative, 3 stars as neutral, and 4–5 stars as positive. While the full dataset contains approximately 700,000 samples, we randomly selected and utilized 100,000 samples for our analysis.

**Emotions.** The Emotion dataset is designed for emotion classification in short text sentences and comprises a total of six classes. It contains approximately 42,000 samples in total. For our experiments, 90% of the data was used for training and the remaining 10% was used for testing.

**SAMSum.** The SAMSum dataset is designed for summarizing conversations in a messenger-style format and contains approximately 16,939 samples. Each entry consists of a conversation paired with its corresponding summary. Unlike traditional text summarization datasets, such as those based on news articles, the SAMSum dataset is characterized by the need to summarize informal, collo-quial dialogues, reflecting more natural conversational patterns.

**SQuAD v1.1** The SQuAD v1.1 is a dataset designed for Machine Reading Comprehension, consist-ing of given contexts paired with corresponding question–answer sets. It is primarily used to train Extractive QA models, with the distinctive feature that the answer to every question is guaranteed to appear within the passage. The dataset contains about 100,000 examples in total, with roughly 88,000 allocated for training and about 10,600 for validation.

## G.2 NETWORK ARCHITECTURES

In this study, a variety of neural network models were utilized to evaluate the Isotropic Gaussian learning. The models used include: Vision Transfomer (ViT), BERT, BERT-Large, GPT-2 Small, MobileNetV2, ResNet-18, and an 8-layer model with only fully connected layers. The details of each model are as follows.

**ViT, BERT, BERT-Large, and GPT-2 Small.** These models were used with their original archi-tectures without structural modifications.

**MobileNetV2 and ResNet-18.** For the convolution filter experiment, the models were kept in their original structure. For the classifier experiment, the original classifier of both models was replaced with a two-layer fully connected classifier, including a ReLU activation function between the layers. While the hidden layer was set to 32 in the main text, experiments were also conducted with various other sizes.

**An 8-layer model with only fully connected layers.** This model is composed entirely of fully connected layers. The size of the hidden units is set to 128. The input is first flattened to match the input dimension of the first layer. A ReLU activation function is applied after each layer, except for the final layer, where a log_softmax function is used for classification.

## G.3 HYPERPARAMETERS

We refer to an 8-layer model with only fully connected layers as 8FFN. Details are mentioned in Table 9.

Table 9: Hyperparameters for each model

| Category | Description | Hyperparameter |
|---|---|---|
| Batch size | For 8FFN, MobileNetV2, ResNet-18, ViT | 32 |
| | For BERT, BERT-Large, GPT-2 Small | 16 |
| Optimizer | For 8FFN | SGD |
| | For MobileNetV2, ResNet-18, ViT, BERT, BERT-Large, GPT-2 Small | AdamW |
| Sampling number | For 8FFN | 100 |
| | For MobileNetV2, ResNet-18, ViT, BERT, BERT-Large, GPT-2 Small | 10 |
| Learning rate | For 8FFN | 1e-3 |
| | For MobileNetV2, ResNet-18 | 1e-4 |
| | For ViT | 5e-5 |
| | For BERT, GPT-2 Small | 2e-5 |
| | For BERT-Large | 3e-5 |
| Weight decay | For 8FFN | 0 |
| | For MobileNetV2, ResNet-18, GPT-2 Small | 0.01 |
| | For ViT, BERT, BERT-Large | 0.001 |
| Split ratio | For Emotions dataset | 0.1 |
| Max length | For BERT input | 128 |
| | For GPT-2 Small | 1024 |
| | For BERT-Large | 384 |
| Sample size | For Yelp Reviews | 100k |

Table 10: The performance of ViT fine-tuned on mini-ImageNet. The 'base' refers to the model trained using back-propagation, and the bold numbers indicate the encoder layer with the Isotropic Gaussian applied.

| layers | base | 1 | 2 | 3 | 4 | 5 | 6 | 7 | 8 | 9 |
|---|---|---|---|---|---|---|---|---|---|---|
| performance(%) | 78.46 | 63.53 | 75.47 | 76.61 | 76.77 | 76.14 | 75.95 | 74.89 | 74.79 | 74.49 |
| layers | 10 | 11 | 12 | 3,7 | 3,12 | 5,6 | 5,7 | 5,12 | 6,7 | 6,12 |
| performance(%) | 72.54 | 72.3 | 75.79 | 73.84 | 73.43 | 72.31 | 72.95 | 73.09 | 71.93 | 73.72 |
| layers | 7,12 | 8,12 | 3,6,9 | 3,7,11 | 4,7,10 | 4,8,12 | 5,9,12 | 6,8,12 | 6,9,12 | 6,10,12 |
| performance(%) | 73.04 | 72.42 | 68.64 | 68.90 | 67.57 | 70.78 | 68.67 | 69.63 | 68.76 | 68.87 |

Table 11: The performance of ViT fine-tuned on Food-101. The 'base' refers to the model trained using back-propagation, and the bold numbers indicate the encoder layer with the Isotropic Gaussian applied.

| layers | base | 1 | 2 | 3 | 4 | 5 | 6 | 7 | 8 |
|---|---|---|---|---|---|---|---|---|---|
| performance(%) | 84.67 | 75.01 | 82.48 | 82.99 | 83.98 | 83.66 | 83.87 | 83.66 | 83.99 |
| layers | 9 | 10 | 11 | 12 | 2,3 | 2,5 | 2,6 | 2,9 | 2,11 |
| performance(%) | 83.41 | 82.99 | 83.16 | 83.87 | 78.31 | 81.00 | 81.01 | 80.64 | 80.05 |
| layers | 3,5 | 3,6 | 3,9 | 3,11 | 3,12 | 4,12 | 5,9 | 5,11 | 5,12 |
| performance(%) | 81.79 | 81.72 | 81.85 | 80.57 | 82.08 | 82.82 | 81.87 | 81.67 | 82.33 |
| layers | 6,9 | 6,11 | 6,12 | 7,12 | 8,12 | 3,7,11 | 4,8,12 | 8,10,12 | 3,6,9,12 |
| performance(%) | 81.57 | 80.88 | 82.65 | 82.32 | 82.56 | 77.04 | 80.44 | 79.46 | 77.23 |

Table 12: The performance of BERT fine-tuned on AG News. The 'base' refers to the model trained using back-propagation, and the bold numbers indicate the encoder layer with the Isotropic Gaussian applied.

| layers | base | 1 | 2 | 3 | 4 | 5 | 6 | 7 | 8 |
|---|---|---|---|---|---|---|---|---|---|
| performance (%) | 94.79 | 94.39 | 94.41 | 94.49 | 94.56 | 94.34 | 94.52 | 94.53 | 94.53 |
| layers | 9 | 10 | 11 | 12 | 5,12 | 6,12 | 7,12 | 8,12 | 9,12 |
| performance (%) | 94.27 | 94.36 | 94.53 | 94.33 | 94.30 | 94.33 | 94.42 | 94.31 | 94.38 |
| layers | 3,5,7 | 4,6,8 | 5,7,9 | 6,8,10 | 4,6,8,10 | 4,8,10,12 | 2,4,6,8,10 | 2,4,6,8,10,12 | 2,4,5,6,8,10,12 |
| performance (%) | 94.02 | 94.16 | 94.22 | 94.30 | 94.07 | 94.14 | 93.98 | 93.69 | 93.61 |

Table 13: The performance of BERT fine-tuned on Yelp Review. The 'base' refers to the model trained using back-propagation, and the bold numbers indicate the encoder layer with the Isotropic Gaussian applied.

| layers | base | 1 | 2 | 3 | 4 | 5 | 6 | 7 | 8 | 9 |
|---|---|---|---|---|---|---|---|---|---|---|
| performance (%) | 88.78 | 88.23 | 88.20 | 88.38 | 88.18 | 88.25 | 88.21 | 88.28 | 88.04 | 88.18 |
| layers | 10 | 11 | 12 | 4,12 | 5,10 | 5,12 | 6,10 | 6,12 | 7,9 | 7,12 |
| performance (%) | 88.10 | 88.28 | 88.33 | 88.07 | 87.86 | 87.98 | 87.67 | 87.86 | 87.95 | 88.12 |
| layers | 8,10 | 8,12 | 9,12 | 4,6,8 | 5,7,9 | 4,6,8,10 | 4,8,10,12 | 2,4,6,8,10 | 2,4,6,8,10,12 | 2,4,5,6,8,10,12 |
| performance (%) | 87.59 | 87.80 | 87.84 | 87.55 | 87.41 | 87.32 | 87.30 | 86.82 | 86.94 | 86.28 |

# H COMPREHENSIVE EXPERIMENT RESULTS

## H.1 TRANSFORMERS

**Vision Transformers (ViT).** Table 10 and 11 present the results of fine-tuning a pretrained ViT model on the mini-ImageNet and Food-101 datasets. The experimental results show that, although performance varies depending on the combination of encoder layers to which the method is applied, overall, the performance remains sufficiently competitive.

**Language models.** Table 12, 13, and 14 present the fine-tuning results of the BERT model on the AG News, Yelp Review, and Emotion datasets, respectively. These experiments confirmed that the methodology also performs well on text-based data. Table 15 presents the results of the text summarization task using GPT-2 Small and the SAMSum dataset. Table 16 shows the results of the experiment using BERT-Large and SQuAD v1.1. The experimental results indicate that, the overall performance was similar to the base performance in the beginning, but the performance gradually decreased as the applied layer increased.

Table 14: The performance of BERT fine-tuned on Emotions. The 'base' refers to the model trained using back-propagation, and the bold numbers indicate the encoder layer with the Isotropic Gaussian applied.

| layers | base | 1 | 2 | 3 | 4 | 5 | 6 | 7 | 8 |
|---|---|---|---|---|---|---|---|---|---|
| performance (%) | 94.29 | 94.13 | 94.24 | 94.14 | 94.13 | 94.12 | 94.13 | 94.08 | 94.11 |
| **layers** | **9** | **10** | **11** | **12** | **4,9** | **4,12** | **5,12** | **6,12** | **8,12** |
| performance (%) | 94.16 | 94.12 | 94.11 | 94.08 | 94.19 | 94.11 | 94.11 | 94.12 | 94.12 |
| **layers** | **4,6,8** | **5,7,9** | **6,8,10** | **8,10,12** | **4,6,8,10** | **4,8,10,12** | **2,4,6,8,10** | **2,4,6,8,10,12** | **2,4,5,6,8,10,12** |
| performance (%) | 94.10 | 94.02 | 94.12 | 94.04 | 94.09 | 94.08 | 94.04 | 93.99 | 93.99 |

Table 15: The performance of GPT-2 Small fine-tuned on SAMSum dataset. The 'base' refers to the model trained using back-propagation, and the bold numbers indicate the decoder layer with the Isotropic Gaussian applied.

| decoder layers | base | 5 | 6 | 7 | 8 | 9 | 10 | 11 | 12 | 7,9 |
|---|---|---|---|---|---|---|---|---|---|---|
| BLEU | 0.6957 | 0.6698 | 0.6724 | 0.6814 | 0.6866 | 0.7006 | 0.6969 | 0.7025 | 0.6963 | 0.6681 |
| ROUGE-1 | 0.7466 | 0.7171 | 0.7116 | 0.7104 | 0.7213 | 0.7291 | 0.7315 | 0.7402 | 0.7404 | 0.6935 |
| ROUGE-L | 0.7419 | 0.7127 | 0.7066 | 0.7052 | 0.7157 | 0.7229 | 0.7246 | 0.7316 | 0.7327 | 0.6897 |
| **decoder layers** | **7,10** | **8,12** | **9,12** | **10,12** | **3,6,9** | **6,8,10** | **6,9,12** | **3,5,7,9** | **4,6,8,10** | **6,8,10,12** |
| BLEU | 0.6692 | 0.6902 | 0.6941 | 0.6935 | 0.6477 | 0.6509 | 0.6505 | 0.6394 | 0.6685 | 0.6457 |
| ROUGE-1 | 0.7045 | 0.7274 | 0.7321 | 0.7287 | 0.6665 | 0.6701 | 0.6683 | 0.6570 | 0.6653 | 0.6569 |
| ROUGE-L | 0.7008 | 0.7218 | 0.7269 | 0.7226 | 0.6606 | 0.6637 | 0.6614 | 0.6509 | 0.6603 | 0.6510 |

Table 16: The performance of BERT-Large fine-tuned on SQuAD v1.1. The 'base' refers to the model trained using back-propagation, and the bold numbers indicate the encoder layer with the Isotropic Gaussian applied.

| encoder layers | base | 1 | 2 | 3 | 4 |
|---|---|---|---|---|---|
| Exact Match (EM) | 84.42 | 83.00 | 82.87 | 82.10 | 82.23 |
| F1 | 90.93 | 89.97 | 89.96 | 89.59 | 89.61 |
| **encoder layers** | **5** | **6** | **7** | **8** | **9** |
| Exact Match (EM) | 81.76 | 81.91 | 81.98 | 82.29 | 81.37 |
| F1 | 89.21 | 89.31 | 89.36 | 89.50 | 88.89 |
| **encoder layers** | **10** | **11** | **12** | **13** | **14** |
| Exact Match (EM) | 82.55 | 80.74 | 81.19 | 82.22 | 80.5 |
| F1 | 89.72 | 88.61 | 88.55 | 89.16 | 88.49 |
| **encoder layers** | **15** | **16** | **17** | **18** | **19** |
| Exact Match (EM) | 81.28 | 81.17 | 82.07 | 82.50 | 82.84 |
| F1 | 88.67 | 88.48 | 89.05 | 89.23 | 89.55 |
| **encoder layers** | **20** | **21** | **22** | **23** | **24** |
| Exact Match (EM) | 82.01 | 83.04 | 83.47 | 83.61 | 83.63 |
| F1 | 89.36 | 89.95 | 90.29 | 90.37 | 90.45 |
| **encoder layers** | **5,17** | **3,11,20** | **4,13,16,22** | **4,13,16,17,22** | **2,6,10,14,18,22** |
| Exact Match (EM) | 82.24 | 82.96 | 80.25 | 74.83 | 79.06 |
| F1 | 89.22 | 89.79 | 87.54 | 83.58 | 87.04 |
| **encoder layers** | **3,7,12, 15,18,20** | **7,10,15, 17,21,23** | **3,6,9,12, 15,18,21** | **1,4,7,9, 12,14,17,19** | **1,4,7,9,11, 12,14,17,19** |
| Exact Match (EM) | 78.31 | 78.48 | 73.32 | 75.57 | 59.76 |
| F1 | 85.86 | 86.66 | 82.31 | 83.78 | 71.97 |
| **encoder layers** | **1,4,7,9,12, 14,15,17,19** | **2,4,7,10,13, 16,18,21,23** | **1,3,5,7,9, 11,13,15,17,19** | **1,4,7,9,10, 12,14,15,17,19** | **2,4,6,8,10, 12,14,16,18,20** |
| Exact Match (EM) | 67.39 | 51.20 | 40.79 | 57.97 | 50.98 |
| F1 | 77.20 | 64.53 | 54.13 | 69.59 | 62.95 |

## H.2 CONVOLUTION NEURAL NETWORKS

Table 17 and 18 present the experimental results for the CNN-based models MobileNetV2 and ResNet-18 on the CIFAR-10 and SVHN datasets. In these experiments, modifications were made to the classifier structure by adding an additional fully connected layer and adjusting the hidden layer sizes.

Table 19 and 20 presnet the experimental results for MobileNetV2 on the CIFAR-10 and SVHN datasets. Table 21 and 22 presnet the experimental results for ResNet-18 on the CIFAR-10 and SVHN datasets. As a result of the experiments, when Isotropic Gaussian learning was applied to a limited number of convolution modules or layers, the performance remained nearly identical to that of the original model. However, once the number of applied modules exceeded a certain threshold, performance began to show a gradual decline.

Table 17: The performance of MobileNetV2 fine-tuned on CIFAR-10 and SVHN. The 'base' refers to the model trained using back-propagation with 1 classifier, and 'Isotropic' denotes the model with 2 classifiers trained with an Isotropic Gaussian applied to the first classifier. The number of parameters refers specifically to those within the classifier for each respective case.

| method | length of hidden layer | CIFAR-10 | | SVHN | | number of parameters |
|---|---|---|---|---|---|---|
| | | train acc (%) | test acc (%) | train acc (%) | test acc (%) | |
| baseline | | 99.28 | 94.87 | 99.70 | 96.42 | 12.81K |
| Isotropic | 256 | 95.68 | 91.37 | 93.38 | 92.81 | 5898 |
| | 128 | 95.68 | 91.37 | 95.00 | 93.97 | 4234 |
| | 64 | 94.56 | 90.77 | 94.58 | 93.25 | 3402 |
| | 32 | 97.09 | 91.68 | 95.52 | 93.73 | 2986 |

Table 18: The performance of ResNet-18 fine-tuned on CIFAR-10 and SVHN. The 'base' refers to the model trained using back-propagation with 1 classifier, and 'Isotropic' denotes the model with 2 classifiers trained with an Isotropic Gaussian applied to the first classifier. The number of parameters refers specifically to those within the classifier for each respective case.

| method | length of hidden layer | CIFAR-10 | | SVHN | | number of parameters |
|---|---|---|---|---|---|---|
| | | train acc (%) | test acc (%) | train acc (%) | test acc (%) | |
| baseline | | 99.17 | 94.47 | 99.60 | 96.57 | 5.13K |
| Isotropic | 128 | 99.04 | 89.87 | 97.38 | 93.43 | 2698 |
| | 64 | 97.36 | 90.13 | 97.92 | 93.42 | 1866 |
| | 32 | 98.46 | 90.23 | 98.88 | 94.04 | 1450 |
| | 16 | 97.85 | 88.67 | 93.20 | 91.94 | 1242 |

## H.3 Fully connected networks

Table 23 presents the experimental results for the 8FFN model with a hidden layer size of 128 on the MNIST dataset. In the experiment, applying the Isotropic Gaussian to the first layer resulted in a significant decrease in performance, and a noticeable performance drop was also observed when applied to the last layer. However, for the other layers, the performance remained sufficiently close to the baseline, demonstrating the robustness of the methodology when carefully applied to intermediate layers.

## I Expanded review of related work

**Distributional approximation of matrix.** The distributional approximation of matrices plays a key role in various fields such as statistics, probability theory, and machine learning. Understanding the distributional properties of matrix-valued statistics is crucial for both theoretical advancement and practical applications, particularly in the analysis of high-dimensional data and complex systems. The matrix Bernstein inequality (Tropp, 2012) provides sharp bounds on the tail probability of the maximum eigenvalue of the sum of independent random matrices. The Bootstrap method is often used for distributional approximation of matrix-valued statistics, as a resampling technique to evaluate the variability of a sample (Fan et al., 2013). In approximation of matrix functions, (Al-Mohy & Higham, 2011) developed an efficient algorithm for computing the matrix exponential function. A shrinkage estimator for covariance matrices in high-dimensional settings (Ledoit & Wolf, 2004) improves the stability and accuracy of the estimation, and (Johnstone, 2001) studied the distribution of the largest eigenvalue in principal component analysis (PCA), enabling significant component testing in high-dimensional data. Prior studies have predominantly focused on approximating entire datasets or systems, aiming to reduce large and complex data. In contrast, our method imporves the specificity of the weight matrix by approximating it on a row-by-row and column-by-column

Table 19: The performance of MobileNetV2 fine-tuned on CIFAR-10. The 'base' refers to the model trained using back-propagation, and the bold numbers indicate the convolution modules with the Isotropic Gaussian applied. The original parameters refer to the parameters of the modules before applying Isotropic Gaussian learning, while the modified parameters are those after applying it. The retention ratio is defined as the ratio of modified parameters to original parameters.

| | performance(%) | Original parameters | Modified parameters | Retention ratio |
|---|---|---|---|---|
| **base** | 94.94 | | | |
| **1** | 92.17 | 864 | 118 | 13.66% |
| **2** | 85.00 | 800 | 178 | 22.25% |
| **3** | 90.95 | 4,704 | 674 | 14.33% |
| **4** | 94.81 | 8,208 | 978 | 11.92% |
| **5** | 89.76 | 9,360 | 994 | 10.62% |
| **6** | 94.93 | 14,016 | 1,298 | 9.26% |
| **7** | 94.78 | 14,016 | 1,298 | 9.26% |
| **8** | 79.63 | 20,160 | 1,362 | 6.76% |
| **9** | 94.79 | 52,608 | 2,578 | 4.90% |
| **10** | 94.78 | 52,608 | 2,578 | 4.90% |
| **11** | 95.14 | 52,608 | 2,578 | 4.90% |
| **12** | 71.75 | 64,896 | 2,642 | 4.07% |
| **13** | 94.66 | 115,776 | 3,858 | 3.33% |
| **14** | 94.71 | 115,776 | 3,858 | 3.33% |
| **15** | 35.36 | 152,640 | 3,986 | 2.61% |
| **16** | 94.59 | 315,840 | 6,418 | 2.03% |
| **17** | 94.84 | 315,840 | 6,418 | 2.03% |
| **18** | 32.05 | 469,440 | 6,738 | 1.44% |
| **19** | 80.78 | 409,600 | 3,200 | 0.78% |
| **4,11** | 93.81 | 60,816 | 3,556 | 5.85% |
| **1,7,16** | 91.20 | 339,264 | 7,834 | 2.31% |
| **1,6,9,14** | 90.99 | 191,808 | 7,852 | 4.09% |
| **4,7,10,13,17** | 93.67 | 506,448 | 15,130 | 2.99% |
| **1,4,6,9,11,13** | 87.28 | 252,624 | 11,408 | 4.52% |
| **4,6,9,11,13,16,17** | 83.20 | 874,896 | 24,168 | 2.76% |
| **1,6,7,9,11,13,14,17** | 86.81 | 690,048 | 22,004 | 3.19% |
| **1,4,6,7,9,10,13,14,16** | 82.86 | 698,256 | 22,922 | 3.28% |
| **1,4,6,7,9,10,11,14,16,17** | 76.48 | 950,928 | 28,162 | 2.96% |

basis. This approach not only reduces computational complexity but also more effectively ensures the retention of inherent characteristics during sampling.

**Random projection.** Gaussian random projection (Bingham & Mannila, 2001) is a technique popularly used for dimensionality reduction (Achlioptas, 2001). Based on the Johnson-Lindenstrauss lemma (Johnson et al., 1986), it aims to preserve the original data structure by maintaining the orthogonality and normality of the rows in the random projection matrix. However, in the context of neural networks, the weight distribution in a fully connected layer should not only reduce dimensionality but also achieve the task objective, working in conjunction with other layers. We finds that random projection is insufficient to achieve this, manifesting the necessity of distribution learning. The proposed method can also be interpreted as projecting large datasets into a more compact representation consisting only of the mean and variance. However, the method differs fundamentally in its approach. Rather than directly reducing the data, it derives the Gaussian characteristics from the data itself, with the key aspect being the generation of a random matrix that follows the Gaussian distribution. More importantly, the proposed method randomly samples a set of weight parameters from the learned Isotropic Gaussian distribution at run-time, in contrast to existing random projection techniques where these weight parameters are typically maintained as fixed values.

**Random matrix.** Random matrices have been applied to neural networks and deep learning, e.g., random matrices enable the efficient transfer of hyper-parameter tuning without the need for network retraining (Yang et al., 2021). While those studies focus on the loss surface (Pennington & Bahri, 2017) and hyperparameter tuning using random matrix theory, none of them have explored learning optimal weight distributions that can fully replace those in fully connected layers. Within computational neuroscience, random matrices have been employed to model synaptic connectiv-

Table 20: The performance of MobileNetV2 fine-tuned on SVHN. The 'base' refers to the model trained using back-propagation, and the bold numbers indicate the convolution modules with the Isotropic Gaussian applied. The original parameters refer to the parameters of the modules before applying Isotropic Gaussian learning, while the modified parameters are those after applying it. The retention ratio is defined as the ratio of modified parameters to original parameters.

| | performance(%) | Original parameters | Modified parameters | Retention ratio |
|---|---|---|---|---|
| **base** | 96.58 | | | |
| **1** | 96.47 | 864 | 118 | 13.66% |
| **2** | 95.60 | 800 | 178 | 22.25% |
| **3** | 96.48 | 4,704 | 674 | 14.33% |
| **4** | 96.54 | 8,208 | 978 | 11.92% |
| **5** | 96.55 | 9,360 | 994 | 10.62% |
| **6** | 96.42 | 14,016 | 1,298 | 9.26% |
| **7** | 96.53 | 14,016 | 1,298 | 9.26% |
| **8** | 95.91 | 20,160 | 1,362 | 6.76% |
| **9** | 96.28 | 52,608 | 2,578 | 4.90% |
| **10** | 96.44 | 52,608 | 2,578 | 4.90% |
| **11** | 96.43 | 52,608 | 2,578 | 4.90% |
| **12** | 92.31 | 64,896 | 2,642 | 4.07% |
| **13** | 96.48 | 115,776 | 3,858 | 3.33% |
| **14** | 96.27 | 115,776 | 3,858 | 3.33% |
| **15** | 90.09 | 152,640 | 3,986 | 2.61% |
| **16** | 96.15 | 315,840 | 6,418 | 2.03% |
| **17** | 96.12 | 315,840 | 6,418 | 2.03% |
| **18** | 94.87 | 469,440 | 6,738 | 1.44% |
| **19** | 95.44 | 409,600 | 3,200 | 0.78% |
| **9,14** | 95.73 | 168,384 | 6,436 | 3.82% |
| **6,13,19** | 94.92 | 539,392 | 8,356 | 1.55% |
| **2,8,11,16** | 95.63 | 389,408 | 10,536 | 2.71% |
| **3,5,7,10 18** | 94.25 | 550,128 | 12,282 | 2.23% |
| **1,4,11,14,17,19** | 94.29 | 902,896 | 17,150 | 1.90% |
| **2,5,9,10,12,15,17** | 79.82 | 648,752 | 19,374 | 2.99% |
| **3,6,7,10,11,14,17,19** | 92.71 | 979,168 | 21,904 | 2.24% |
| **1,3,5,7,12,13,16,17,19** | 76.35 | 1,250,896 | 25,220 | 2.02% |
| **3,6,7,9,10,11,14,16,17,19** | 92.71 | 1,357,616 | 30,840 | 2.27% |

ity between neurons in the brain (Sompolinsky et al., 1988; Rajan & Abbott, 2006; Timme et al., 2004). However, given that these studies primarily aim to develop new brain-inspired models, their approaches are not directly applicable to the fully connected neural network architecture, unlike the proposed method.

**Weight initialization.** The proposed method can be regarded as a form of run-time weight initialization (sampling), similar to Xavier (Glorot & Bengio, 2010) and He (He et al., 2015) initialization (sampling), as it randomly initializes (samples) weight parameters from the learned distribution each time the layer is executed. While existing weight initialization methods initialize weight parameters before model training to maintain consistent standard deviations across layers—by setting the mean to zero and the variance as a function of the layer's fan-in and fan-out—they still require the initialized weight parameters to be learned (optimized) through training to achieve the task objective. Thus, simply sampling (initializing) weight parameters using those weight initialization methods during forward propagation, without distribution learning, is insufficient for achieving optimal model performance.

Table 21: The performance of ResNet-18 fine-tuned on CIFAR-10. The 'base' refers to the model trained using back-propagation, and the bold numbers indicate the convolution modules with the Isotropic Gaussian applied. The original parameters refer to the parameters of the modules before applying Isotropic Gaussian learning, while the modified parameters are those after applying it. The retention ratio is defined as the ratio of modified parameters to original parameters.

| | performance(%) | Original parameters | Modified parameters | Retention ratio |
|---|---|---|---|---|
| base | 94.64 | | | |
| 1 | 91.47 | 9,408 | 422 | 4.49% |
| 2 | 94.38 | 36,864 | 1,280 | 3.47% |
| 3 | 93.99 | 36,864 | 1,280 | 3.47% |
| 4 | 93.88 | 36,864 | 1,280 | 3.47% |
| 5 | 94.45 | 36,864 | 1,280 | 3.47% |
| 6 | 93.60 | 73,728 | 1,408 | 1.91% |
| 7 | 89.52 | 147,456 | 2,560 | 1.74% |
| 8 | 94.78 | 8,192 | 384 | 4.69% |
| 9 | 94.85 | 147,456 | 2,560 | 1.74% |
| 10 | 94.57 | 147,456 | 2,560 | 1.74% |
| 11 | 93.81 | 294,912 | 2,816 | 0.95% |
| 12 | 93.77 | 589,824 | 5,120 | 0.87% |
| 13 | 94.34 | 32,768 | 768 | 2.34% |
| 14 | 94.70 | 589,824 | 5,120 | 0.87% |
| 15 | 94.56 | 589,824 | 5,120 | 0.87% |
| 16 | 93.70 | 1,179,648 | 5,632 | 0.48% |
| 17 | 93.36 | 2,359,296 | 10,240 | 0.43% |
| 18 | 94.24 | 131,072 | 1,536 | 1.17% |
| 19 | 90.36 | 2,359,296 | 10,240 | 0.43% |
| 20 | 94.13 | 2,359,296 | 10,240 | 0.43% |
| 10,16 | 92.60 | 1,327,104 | 8,192 | 0.62% |
| 4,7,13 | 93.41 | 217,088 | 4,608 | 2.12% |
| 2,5,10,16 | 90.22 | 1,400,832 | 10,752 | 0.77% |
| 3,4,6,9,13 | 89.89 | 327,680 | 7,296 | 2.23% |
| 2,5,8,10,13,16 | 89.43 | 1,441,792 | 11,904 | 0.83% |
| 3,4,6,9,13,15,18 | 85.25 | 1,048,576 | 13,952 | 1.33% |
| 2,4,5,6,9,11,13,15 | 82.19 | 1,247,280 | 16,512 | 1.32% |
| 2,3,5,6,9,10,13,15,18 | 82.05 | 1,232,896 | 19,792 | 1.61% |
| 2,3,5,6,9,10,13,14,15,18 | 83.65 | 1,826,720 | 22,912 | 1.25% |

Table 22: The performance of ResNet-18 fine-tuned on SVHN. The 'base' refers to the model trained using back-propagation, and the bold numbers indicate the convolution modules with the Isotropic Gaussian applied. The original parameters refer to the parameters of the modules before applying Isotropic Gaussian learning, while the modified parameters are those after applying it. The retention ratio is defined as the ratio of modified parameters to original parameters.

| | performance(%) | Original parameters | Modified parameters | Retention ratio |
|---|---|---|---|---|
| **base** | 96.57 | | | |
| **1** | 96.54 | 9,408 | 422 | 4.49% |
| **2** | 96.37 | 36,864 | 1,280 | 3.47% |
| **3** | 96.36 | 36,864 | 1,280 | 3.47% |
| **4** | 96.15 | 36,864 | 1,280 | 3.47% |
| **5** | 96.69 | 36,864 | 1,280 | 3.47% |
| **6** | 96.55 | 73,728 | 1,408 | 1.91% |
| **7** | 96.40 | 147,456 | 2,560 | 1.74% |
| **8** | 96.58 | 8,192 | 384 | 4.69% |
| **9** | 96.45 | 147,456 | 2,560 | 1.74% |
| **10** | 96.77 | 147,456 | 2,560 | 1.74% |
| **11** | 96.42 | 294,912 | 2,816 | 0.95% |
| **12** | 96.50 | 589,824 | 5,120 | 0.87% |
| **13** | 96.41 | 32,768 | 768 | 2.34% |
| **14** | 96.39 | 589,824 | 5,120 | 0.87% |
| **15** | 96.32 | 589,824 | 5,120 | 0.87% |
| **16** | 95.99 | 1,179,648 | 5,632 | 0.48% |
| **17** | 96.15 | 2,359,296 | 10,240 | 0.43% |
| **18** | 96.40 | 131,072 | 1,536 | 1.17% |
| **19** | 95.06 | 2,359,296 | 10,240 | 0.43% |
| **20** | 96.48 | 2,359,296 | 10,240 | 0.43% |
| **1,15** | 96.27 | 599,232 | 5,542 | 0.92% |
| **6,13,18** | 96.34 | 237,568 | 3,712 | 1.56% |
| **2,8,11,18** | 96.07 | 471,040 | 6,016 | 1.28% |
| **4,10,12,17,19** | 90.59 | 5,492,736 | 29,440 | 0.54% |
| **1,6,9,12,15,18** | 91.10 | 1,541,312 | 16,166 | 1.05% |
| **1,7,10,11,13,15,18** | 91.95 | 1,352,896 | 15,782 | 1.17% |
| **2,4,5,6,11,12,17,18** | 88.92 | 3,565,424 | 25,960 | 0.73% |
| **2,4,6,8,10,12,14,16,18** | 82.39 | 2,797,472 | 24,320 | 0.87% |
| **2,4,5,6,8,11,13,15,17,18** | 82.63 | 3,600,304 | 26,112 | 0.73% |

Table 23: The train and test accuracy of 8-layer networks on MNIST consisting of 128 neurons per hidden layer. Here, 'base' refers to back-propagation training, and the bold numbers in the table indicate the layers applying the Isotropic Gaussian.

| layers | base | 1 | 2 | 3 | 4 | 5 | 6 | 7 | 8 | 3,5 | 3,6 | 3,7 |
|---|---|---|---|---|---|---|---|---|---|---|---|---|
| train acc(%) | 99.94 | 22.65 | 89.29 | 97.45 | 97.45 | 96.46 | 94.89 | 97.14 | 26.31 | 96.44 | 96.72 | 97.1 |
| test acc(%) | 95.81 | 22.51 | 83.64 | 94.79 | 94.79 | 95.62 | 94.84 | 96.76 | 24.76 | 94.38 | 94.87 | 93.85 |
| **layers** | **4,5** | **4,6** | **4,7** | **5,6** | **5,7** | **6,7** | **3,4,5** | **3,5,7** | **4,5,6** | **3,4,5,6** | **4,5,6,7** | **3,4,5,6,7** |
| train acc(%) | 93.52 | 96.58 | 98.75 | 92.46 | 99.11 | 96.07 | 93.82 | 96.85 | 93.75 | 93.49 | 96.76 | 87.45 |
| test acc(%) | 94.08 | 95.60 | 96.43 | 93.16 | 97.39 | 95.32 | 92.93 | 93.53 | 88.99 | 93.15 | 95.78 | 85.37 |

