# OpenReview forum: "An Isotropic Gaussian Perspective on Fully Connected Layers"
_ICLR.cc/2026/Conference — ICLR 2026 Conference Withdrawn Submission_

### Official Review · Reviewer_pZUA · 2025-10-30

**Soundness:** 3
**Presentation:** 2
**Contribution:** 3
**Rating:** 4
**Confidence:** 3

**Summary:**

This paper introduces an Isotropic Gaussian Parameterization framework that represents fully-connected layer weights as compositions of row- and column-wise isotropic Gaussian distributions. By learning the mean and variance of both row and column components, the method reconstructs weight matrices through a differentiable Gaussian merging function, significantly reducing parameter count from $O(nm)$ to $O(n+m)$. Two optimization schemes, gradient distribution update and weight distribution update, are proposed to learn these distributions efficiently. Extensive experiments on diverse architectures demonstrate remarkable compression with minimal performance degradation.

**Strengths:**

1.Ingenious Formulation: The construction of Equation (2) is a significant contribution. It compresses the weight matrix parameters from $\mathcal{O}(nm)$ to $\mathcal{O}(n+m)$ by modeling row-wise and column-wise distributions. This approach theoretically integrates dual-perspective information, offering both high interpretability and substantial parameter savings.

2.Novel Learning Methods: GDU models the gradient distribution, while WDU models the distribution of updated weights. This provides an intuitive illustration of the exploratory growth of the variance of GDU and the stability decline of the variance of WDU. The comparative analysis clearly reveals the different inductive biases of the two methods.

3.Comprehensive Experimentation: The method is validated across diverse backbones. These experiments demonstrate significant parameter reduction while maintaining comparable performance when a limited number of layers are replaced.

**Weaknesses:**

1.Lack of Derivation for Equation (2): In Section 2.2, the paper directly presents the formula for fusing row and column distributions (Equation 2) without providing a theoretical derivation or citing relavant work.

2.Layer-Sensitivity and Applicability Limits: A more thorough analysis of layer-wise sensitivity is needed. The method is not universally applicable across all layers, as evidenced by Table 23. Applying the method to the first or last layer of the 8-layer FFN results in a catastrophic performance collapse (test accuracies of 22.51% and 24.76%, respectively). This indicates that this method is highly unsuitable for the layer that plays a crucial role in feature extraction and classification.

3.Outdated Baselines: The baselines used for comparison in the experiments are relatively dated. A comparison against more recent state-of-the-art compression or efficient architectures is necessary to validate the method's effectiveness.

**Questions:**

Missing Hyperparameter Sensitivity Analysis: While hyperparameter settings are provided, the paper lacks a stability analysis. The studies on key hyperparameters are needed to demonstrate the method's robustness.

---

### Official Review · Reviewer_WEnp · 2025-10-31

**Soundness:** 1
**Presentation:** 2
**Contribution:** 1
**Rating:** 2
**Confidence:** 4

**Summary:**

This paper proposes representing fully connected (FC) layers in neural networks as Isotropic Gaussian distributions, parameterized by per-row and per-column means and variances. The authors introduce two methods for learning these distributions — Gradient Distribution Update (GDU) and Weight Distribution Update (WDU) — which integrate with backpropagation. Once trained, weights are sampled from the learned Isotropic Gaussian distributions. The claimed benefits include: parameter reduction from (O(n^2)) to (O(n)) or (O(n+m)), minimal loss in accuracy across a range of models (ViT, BERT, GPT-2, ResNet, etc.), a new statistical interpretation of FC layers. The paper reports extensive empirical results showing that selectively replacing FC layers with Isotropic Gaussian ones leads to small performance drops while drastically reducing the parameter count.
Overall, the paper introduces an interesting but conceptually shallow and empirically underdeveloped approach. The isotropic Gaussian formulation does not provide compelling theoretical or practical advantages over existing Bayesian or compression techniques. While competently executed, the work falls short of the novelty, rigor, and impact expected. Strengthening theoretical grounding, comparing against stronger baselines, and demonstrating clear computational benefits would be necessary for reconsideration.

**Strengths:**

- The proposed view is conceptually interesting, linking weight compression and probabilistic parameterization.
- The authors conduct a broad set of experiments across architectures and domains.

**Weaknesses:**

- Lack of theoretical justification: The isotropy assumption is overly restrictive and unjustified for real weight distributions. No theoretical analysis is provided to support when or why FC weights can be well-modeled by such simple Gaussians.
- Limited novelty vs. Bayesian and low-rank methods: The idea of learning Gaussian distributions over weights is longstanding (Bayes by Backprop, variational dropout, etc.). The paper’s proposed GDU/WDU updates largely replicate standard moment-matching or reparameterization schemes with superficial differences.
- Empirical results lack depth: Reported performance drops (e.g., 1–3% accuracy) are not rigorously analyzed (no ablation on variance, sampling noise, or computational cost). The claimed parameter reduction ignores that sampling at runtime still requires instantiation of full weight matrices. There is no comparison to more established compression baselines (low-rank factorization, quantization, pruning).
- Questionable practical relevance: The experiments suggest severe degradation when applied to more than a few layers, limiting usefulness. Runtime efficiency is not discussed; memory savings during training/inference are unclear. The method introduces stochasticity without clear performance or interpretability gains.
- Positioning and clarity issues: The paper overstates originality relative to Bayesian NNs and random feature methods. The empirical section is excessively long but lacks analysis or insight beyond tabulated metrics.

**Questions:**

1) How does the proposed approach differ, in substance, from existing Bayesian neural network methods that also model weights as Gaussians?
2) Have you compared your approach to low-rank or tensor decomposition methods, which also achieve O(n)–like compression?
3) The isotropic assumption ignores covariance among weights. Did you test anisotropic or diagonal Gaussian variants to see if they preserve performance better?
4) How sensitive are results to the sampling variance and random seeds? Do you observe instability during training or inference due to stochasticity?
5) Could the proposed GDU/WDU updates be interpreted as special cases of reparametrized gradient updates or variational inference? If so, what new insight do they offer?

---

### Official Review · Reviewer_r6Tk · 2025-10-31

**Soundness:** 1
**Presentation:** 1
**Contribution:** 2
**Rating:** 2
**Confidence:** 4

**Summary:**

This paper proposes to replace a "subset" of the fully-connected layers in the deep learning models with a parameterization modeled by isotropic Gaussian distributions in order to reduce the memory requirements of their weight parameters by reducing the number of trainable parameters. The authors propose two simple distribution learning methods to train the means and variances of the isotropic Gaussian distributions modeling the weights. Ultimately, they rely on extensive experiments to demonstrate the benefits of the proposed technique.

**Strengths:**

1. Interesting and potentially beneficial research question (modeling the parameters of neural networks with Gaussian distributions) that can be utilized for both theory and practice. This can reduce the number of trainable parameters (for easier training in practice) and lead to models that can be theoretically analyzed with the help of Gaussian analysis tools.

2. Having separate column-wise and row-wise statistics (means and variances) and combining them to generate weight matrices sounds like a good idea.

3. Various experiments have been conducted to illustrate the benefits of the proposed technique.

**Weaknesses:**

1. The writing in the paper is informal in some parts, requiring significant improvement to match the presentation level of an ICLR-level paper. For example, in many locations where the isotropic Gaussian is mentioned, the meaning is ambiguous from a mathematical point of view (see Minor 1 below as an example).

2. The proposed way of combining column-wise and row-wise statistics (means and variances) to generate weight matrices (in Equation 2) is not fully motivated. One can understand that this combination is a weighted average (with respect to dimensions) of the statistics, together with a discrepancy penalty for regularization. However, further explanation for the motivation is required.

3. While the paper relies on intuitions to develop the method and relies on the experimental results to confirm them, the experimental results are not enough to confirm the claims. Replacing some layers of a deep model with the proposed technique and demonstrating that the performance drop is small is insufficient to show the claimed benefit of the technique. Indeed, it brings up many questions (see my questions below).

4. Only a subset of layers in the models is changed with the so-called Isotropic Gaussian layers, and how to select those layers is not discussed. So, the understanding of when the proposed approach is viable is limited.

5. In most of the experimental results, we can see a performance drop when the proposed technique is applied. Therefore, there is a clear trade-off between performance and applying the technique to reduce the number of parameters.

6. The paper does not include any mention of the theoretical properties of the proposed technique, although the Gaussian nature of the setting allows many analysis tools. Furthermore, the authors only provide numerical results for the evolution of element-wise means and variances; however, joint statistics (of the elements of the weight matrix) are also relevant and should be studied. Furthermore, the representation ability of the proposed Isotropic Gaussian modeling should be studied.

7. Although it might not be quite apparent at first glance, this work is related to a line of work that studies data modeling with Gaussian distributions and another line of work that considers model parameters as a data modality (one can train another model to generate parameters for the model of interest). These connections are not mentioned in the paper.
   * See "Gaussian splatting" literature for examples of the first one
   * See "ICLR 2025 workshop: Neural Network Weights as a New Data Modality" for examples of the second

Overall, I believe there are several fundamental gaps in this work, and until these are addressed, I recommend rejecting the paper.

**Questions:**

1. Regarding the the proposed way of combining column-wise and row-wise statistics:
   * Why is the weighted average (with respect to dimensions) used?
   * Why is the regularization chosen this way?

2. The authors provided a comparison against alternative distributions. However, a good baseline would be properly initilized and fixed weight, using stardard initialization practices in the literature. Have the authors tried this?

3. Why is the performance drop is big when all the MLP layers are replaced with the Isotropic Gaussian modeling?

4. How can we properly select the subset of layers for the Isotropic Gaussian modeling?

5. How the authors checked the representation ability of the proposed Isotropic Gaussian modeling?
   * a) What is the maximum possible rank of the expected (mean) weight matrix?
   * b) If we vectorize the weight matrix and study its mean and covariance, what kind of structure would we see?

6. Is it possible to avoid performance drops because of such Gaussian modeling of the weight matrices? Have the authors mathematically checked what might be the reason for the performance drops?

**Minor:**

1. Ambiguous sentence in Line 164: "Deriving the mean and variance of Isotropic Gaussian distributions analytically is highly challenging
and often requires unrealistic assumptions." Indeed, the mean and variance of Gaussian distributions are easy to derive analytically, but the authors are referring to something else.

---

### Official Review · Reviewer_D7kE · 2025-11-05

**Soundness:** 2
**Presentation:** 3
**Contribution:** 2
**Rating:** 2
**Confidence:** 5

**Summary:**

Modeling of the weight distribution (Eq.(2)) is interesting. The proposed training methods, i.e., GDU and WDU, are simple and easy to follow. However, some experiments are unclear or incomplete, which makes the results unconvincing. Therefore, I vote for weak/borderline reject.

**Strengths:**

1. It is interesting to model the distribution of the (i, j)-th element of the weight matrix as a Gaussian, whose mean and variance are induced from those of the i-th row-wise Gaussian and the j-th column-wise Gaussian (Eq.(2)).
2. The proposed GDU/WDU are easy to follow and seem to be easily implemented.

**Weaknesses:**

1. The implementations of Eqs.(5)-(6) are confusing. It is unclear which distribution the expectation takes on in Eqs.(5)-(6). According to Eq.(2), the elements of W_{i, :} are not i.i.d. N(\mu_{i, :}, \sigma_{i, :}). Thus, it lacks mathematical explanation to approximate Eqs.(5)-(6) with the mean and variance of the elements in the vector W_{i, :}. In short, Eqs.(5)-(6) seem only heuristic methods to me.
2. There are several unclear/incomplete points in experiments, which make the result unconvincing:
- The provided code lacks implementations of language models such as ViTs, BERT, GPT-2 Small, and BERT-Large, which makes results in Table 1-4 rather unconvincing.
- It is unclear whether the results in Table 1-7 are of GDU or WDU. There should be a comparison of performance between GDU and WDU.
- It is unclear what the “BP(%)” in Table 8 is. In Sec.4.4, what is “backpropagation-based distributions”?
- It is unclear how many epochs or steps are required for GDU/WDU. According to Figure 3, there are at most 500 epochs, which may take a very long time.
- There is no experimental results of SVHN, CIFAR-10, MNIST in Table 1.
- Tables 1-4 only lists the comparisons of number of weights of single layer. It would be better to include the comparisons of number of weights of the whole model.
3. It is tricky to pick which/how many layer(s) to apply Isotropic Gaussian Learning and may require extensive hyperparameter tuning. This may hinder the practicability of the proposed methods.
4. It lacks experiments on the sampling number and running time. According to Table 9, 10 rounds of samplings are needed, which may require a long running time compared to ‘base’ model during inference.
5. Eqs.(8)-(9) lack proofs.

**Questions:**

1. About the second term of \sigma^2_{i, j} in Eq.(2):
- Why choose the coefficient \frac{mn}{(m+n)^2}? Is there a mathematical explanation or it is just heuristic?
- Could the authors provide ablation studies on this term?
- The authors state that ‘This second term serves as an adaptive regularization mechanism, explicitly capturing uncertainty when the two distributions diverge.’. Could the authors explain what would happen if ‘the two distributions diverge’?
2. Could the authors conduct experiments on replacing some fully connected layers with BNN layers (rather than Isotropic Gaussian Layer), with training method like Bayes-by-backprop or variants of the proposed GDU/WDU? It could better demonstrate the representational capability and efficiency of the proposed Isotropic Gaussian Layer.

---

### Note · Authors · 2025-11-19

I have read and agree with the venue's withdrawal policy on behalf of myself and my co-authors.